# Mapping of ESA-CCI land cover data to plant functional types for use in the CLASSIC land model

Libo Wang[1], Vivek K. Arora[2], Paul Bartlett[1], Ed Chan[1], and Salvatore R. Curasi[34]

[1]Climate Processes Section, Climate Research Division, Environment and Climate Change Canada, Toronto, ON, Canada

[2]Canadian Centre for Climate Modelling and Analysis, Climate Research Division, Environment and Climate Change Canada, Victoria, BC, Canada

[3]Department of Geography and Environmental Studies, Carleton University, Ottawa, ON, Canada.

[4]Climate Processes Section, Climate Research Division, Environment and Climate Change Canada, Victoria, BC, Canada.

Corresponding author: Libo Wang, Libo.Wang@ec.gc.ca

**Abstract**
Plant functional types (PFTs) are used to represent vegetation distribution in land surface models
(LSMs). Previous studies have shown large differences in the geographical distribution of PFTs
currently used in various LSMs, which may arise from the differences in the underlying land
cover products but also the methods used to map or reclassify land cover data to the PFTs that a
given LSM represents. There are large uncertainties associated with existing PFT mapping
methods since they are largely based on expert judgment and therefore are subjective. In this
study, we propose a new approach to inform the mapping or the cross-walking process using
analyses from sub-pixel fractional error matrices, which allows for a quantitative assessment of
the fractional composition of the land cover categories in a dataset. We use the Climate Change
Initiative (CCI) land cover product produced by the European Space Agency (ESA). Previous
work has shown that compared to fine-resolution maps over Canada, the ESA-CCI product
provides an improved land cover distribution compared to that from the GLC2000 dataset
currently used in the CLASSIC (Canadian Land Surface Scheme Including Biogeochemical
Cycles) model. A tree cover fraction dataset and a fine-resolution land cover map over Canada
are used to compute the sub-pixel fractional composition of the land cover classes in ESA-CCI,
which is then used to create a cross-walking table for mapping the ESA-CCI land cover
categories to nine PFTs represented in the CLASSIC model. There are large differences between
the new PFT distributions and those currently used in the model. Offline simulations performed
with the CLASSIC model using the ESA-CCI based PFTs show improved winter albedo
compared to that based on the GLC2000 dataset. This emphasizes the importance of accurate
representation of vegetation distribution for realistic simulation of surface albedo in LSMs.
Results in this study suggest that the sub-pixel fractional composition analyses are an effective
way to reduce uncertainties in the PFT mapping process and therefore, to some extent, objectify
the otherwise subjective process.

## 1. Introduction

Land cover is a critical component of the earth system that affects the exchange of energy, water,
and carbon between the land surface and the atmosphere (Pielke et al., 1998; Sterling et al.,
2013). Accurate representation of global land cover (LC) is important for land surface models
(LSMs) which provide the lower boundary conditions to the atmosphere in numerical weather
forecasting, climate, and earth system models (ESMs). Plant functional types (PFTs) are groups
of plant species that share similar structural, phenological, and physiological traits, and have
been commonly used in LSMs to represent vegetation distribution. This simplification has
allowed the simulation of structural attributes of vegetation dynamically within ESMs (Arora &
Boer, 2010; Bonan et al., 2003; Krinner et al., 2005). In order to improve the representation of
ecosystem ecology and vegetation demographic processes within ESMs, both species-based and
trait-based models have been attempted in LSMs (Fisher et al., 2018; Zakharova et al., 2019).
However, these individual-based models are computationally too expensive to model
biogeochemical processes, especially photosynthesis and the carbon cycle at the global scale
(Bonan et al., 2002; Smith et al., 1997; 2001). As a compromise, "cohort-based" models have
been developed where individual plants with similar properties (size, age, functional type) are
grouped together and have been implemented in some ESMs (Fisher et al., 2018). Though there
are limitations in PFTs-based models (Scheiter et al., 2013; Zakharova et al., 2019), PFTs are
commonly used in LSMs that participate routinely in the Global Carbon Project (Friedlingstein
et al., 2020) and in ESMs that participate in the Coupled Models Intercomparison Project (CMIP,
Wang et al., 2016).
There are three approaches for modeling PFTs: (1) static, where the fractional coverage of PFTs
is prescribed and does not vary through time; (2) forced, where the fractional coverage of PFTs
is still prescribed but vary through time based on scenarios of land cover/land-use change; and
(3) dynamic, where the fractional coverage of PFTs is simulated dynamically with competition
for available space and resources between PFTs (Fisher et al., 2018; Melton and Arora, 2016).
The number and type of PFTs used in each LSM differ. Global land cover datasets are typically
used to derive the fractional coverage of PFTs for use in LSMs. However, large differences exist
in both the fractional coverage and the geographical distribution of PFTs, which are caused by
differences in the LC datasets themselves but also due to the methods used to map LC datasets to
the PFTs represented in various models (Fritz et al, 2011; Hartley et al., 2017; Ottle et al., 2013;
Wang et al., 2016).
Since different PFTs are characterized by different physical and biogeochemical processes and
parameter values, the spatial distribution and fractional cover of PFTs constitute one of the
important geophysical fields that are required for realistic simulation of carbon, water, and
energy budgets in LSMs (Arora and Boer, 2010; Betts, 2001). For example, the surface
roughness for short or tall vegetation is very different, which affects simulated turbulent
exchanges. The surface albedos for needleleaf evergreen trees, broadleaf deciduous trees, and
grasslands are also very different, especially during winter when deciduous trees are leafless and
short vegetation is largely buried by snow (Bartlett and Verseghy, 2015; Moody et al., 2007).
Wang et al. (2016) found that the bias in winter albedo in selected boreal forest regions among
the CMIP5 models was largely related to biases in leaf area index (LAI) and tree cover fraction.
Model experiments using the MPI-ESM by Georgievski and Hagemann (2019) suggested that
uncertainties in vegetation distribution may lead to noticeable variations in near-surface climate
variables and large-scale circulation patterns.
The Canadian Land Surface Scheme Including Biogeochemical Cycles (CLASSIC) is an open-
source community land model that is designed to address research questions that explore the role
of the land surface in the global climate system (Melton et al., 2020). It is the successor to the
coupled modelling framework based on the Canadian Land Surface Scheme (CLASS; Verseghy,
1991; 1993) and the Canadian Terrestrial Ecosystem Model (CTEM; Arora and Boer, 2005;
Melton and Arora, 2016). The physics and biogeochemistry modules of CLASSIC are based on
CLASS and CTEM models, respectively. Since the development of CTEM in the early 2000s,
the GLC2000 LC product has been used to specify the spatial distribution of PFTs for CLASSIC
when employed as the land surface component of the Canadian Earth system model developed
by Environment and Climate Change Canada (Arora et al., 2009; Wang et al., 2006). The
Climate Change Initiative (CCI) LC product recently produced by the European Space Agency
(ESA) is available at an annual temporal resolution for the period 1992 to 2018 at 300 m spatial
resolution (ESA, 2017). It was produced based on broad user consultation, specifically to address
the needs of the climate modelling community (Bontemps et al., 2012). Wang et al. (2019)
showed that when compared to the finer resolution maps over Canada, the 300 m ESA-CCI
product provides much improved LC distribution over Canada compared to that from the 1 km
GLC2000 dataset.
To map LC classes to PFTs, a cross-walking table (CW-table) is usually created to assign
fractions of each LC class to the different PFTs, such that the sum of the fractions for each class
is always one (including fractions of water and bare ground). Previous methods for creating such
CW-tables are mainly based on LC class descriptions, expert knowledge, and the spatial
distribution of global biomes (Ottle et al., 2013; Poulter et al., 2011; 2015; Sun and Liang, 2007;
Wang et al., 2006). Because LC maps only provide the types of vegetation, and each class can be
associated with a broad range of fractional cover of either one or more vegetation types, there are
large uncertainties associated with any cross-walking or reclassification process. Wang et al.
(2019) reclassified the 10 PFTs in the default CW-table provided in the ESA-CCI LC product
user manual (Table 7-2, ESA, 2017) into PFTs represented in the CLASSIC model and
compared them with those based on the GLC2000 dataset. The results suggest that uncertainties
in the CW-tables were a major source of large differences in the PFT distributions. In addition,
the fractional coverage of tree PFTs based on the default CW-table for the ESA-CCI LC dataset
was overestimated along the taiga-tundra transition zone in western Canada, which led to
underestimation in winter albedo in CLASSIC offline simulations driven with observation-based
reanalysis data (Wang et al., 2018).
The objective of this study is to develop a new CW-table for reclassifying the ESA-CCI LC
classes into PFTs represented in the CLASSIC model over the model's Canadian domain, and to
compare and assess the performance of CLASSIC offline simulations using the new and existing
PFT distributions. Given the close link between the bias in winter albedo and the vegetation
distribution in the models (Wang et al., 2016), our assessment of model performance focuses on
the simulated surface albedo during the maximum snow accumulation period (February–March
for the boreal forest). This simplifies our analyses by excluding the fall/spring transition periods
when biases in snow accumulation and melt timing can have a large influence on surface albedo
simulated by LSMs (Wang et al., 2014). In addition, we extend the CW-table for the ESA-CCI
LC dataset to the global domain. A comprehensive assessment of the impact of the PFT
distribution based on the new CW-table and the ESA-CCI LC dataset on the performance of the
CLASSIC model at the global scale is presented in Arora et al. (2022).
**2. Data and the CLASSIC model**
**2.1 The Hybrid LC map over Canada**
The United States Geological Survey archive of Landsat imagery has provided open and free
access to georeferenced and spectrally corrected analysis-ready imagery (Wulder et al., 2012),
which makes it possible to generate time series of LC maps to study LC change. Recently two of
these products based on Landsat imagery were generated over Canada, including the North
America Land Change Monitoring System (NALCMS) LC dataset (Latifovic et al., 2017) and
the Virtual Land Cover Engine (VLCE) framework-generated LC dataset (Hermosilla et al.,

125  2018).

Based on the random forest algorithm and local optimization method, the Canada Centre for
Remote Sensing has generated the NALCMS LC maps of Canada for the years 2010 and 2015 at
30 m resolution using Landsat imagery (Latifovic et al., 2017). These LC products are the
Canadian contribution to the 30 m resolution 2010/2015 LC map of North America to the joint
collaborative effort by the Mexican, American, and Canadian government institutions under the
NALCMS umbrella. The NALCMS LC map has 19 classes based on the United Nations Land
Cover Classification System (LCCS; Di Gregorio, 2005). Assessment based on reference
samples showed an overall accuracy of 76.6% for the year 2010 data (Latifovic et al., 2017),
which is used in this study.
VLCE is an automated framework to enable change-informed annual LC mapping using time
series of Landsat surface reflectance. Temporally consistent annual LC maps representative of
Canada's forested ecosystems from 1984 to 2012 were generated using the VLCE framework,
characterizing LC dynamics following wildfire and harvesting events by Hermosilla et al.
(2018). The VLCE maps have 12 LC classes in a hierarchical classification structure following
that of the National Forest Inventory. Assessment based on reference samples showed an overall
accuracy of 70.3% for the map of the year 2005 (the year with the greatest number of reference
samples; Hermosilla et al., 2018). Land cover data from the year 2010 are used in this study.
Overall, the 19-class NALCMS product presents a more detailed LC distribution than the 12-
class VLCE map over Canada. For example, areas classified as "Exposed/Barren lands" in the
VLCE map correspond to either "Sub-polar or polar grassland-lichen-moss", "Sub-polar or polar
barren-lichen-moss", or "Barren lands" in the NALCMS map. Areas of cropland are not
separated from grassland in the VLCE map. A recent study showed that the wetland class in
NALCMS suffers from large uncertainty in forest cover mapping because treed-wetland was not
separated from herbaceous wetland in its legend (Wang et al., 2019). To take advantage of both
datasets, we created a hybrid product by combining them through the following steps: (1)
Reproject the VLCE data from its Lambert Conformal Conic projection to the same Lambert
Azimuthal Equal Area projection that is used for the NALCMS data; (2) Replace pixels
classified as "Exposed/Barren lands" and "Bryoids" in the VLCE data with the more specific LC
classes from the NALCMS data; (3) Replace pixels classified as "Herbs" in the VLCE data with
the "Cropland" class in the NALCMS data (remains "Herbs" if not classified as "Cropland" in
NALCMS); (4) and merge the rest of LC classes from NALCMS to the corresponding classes in
the VLCE data. There are a total of 17 classes in this new hybrid product, which we will
henceforth refer to it as the Hybrid LC dataset and is shown in Figure 1.
**2.2 The global LC products**
The GLC2000 dataset was generated from SPOT/VEG data collected from November 1999 to
December 2000 at 1 km resolution (Bartholomé and Belward, 2005). It was produced by 21
separate regional expert groups using an unsupervised image classification method. Based on the
LCCS, the regional products were merged into one global product with a generalized LCCS
legend of 22 classes. Assessment based on a random sampling of reference sites globally
estimated an overall accuracy of 68.6% for the GLC2000 product (Mayaux et al., 2006).
The annual ESA-CCI LC data at 300 m resolution are available for the period 1992-2018, which
were generated from baseline data and annual LC changes (ESA, 2017). The baseline data were
generated using a combination of machine learning and unsupervised image classification
methods from the entire archive of ENVISAT/Medium Resolution Imaging Spectrometer for the
period of 2003-2012. The annual LC changes were detected at 1 km resolution from the
Advanced Very High Resolution Radiometer time series between 1992 and 1999, SPOT/VEG
time series between 1999 and 2013, and the PROBA-V time series between 2013 and 2018.
Based on the LCCS legend, the ESA-CCI LC data have 22 level 1 classes, and 15 level 2 sub-
classes. Assessment based on the GlobCover validation database estimated an overall accuracy
of 71% for the ESA-CCI LC product (ESA, 2017).
**2.3 Other datasets**
Airborne Lidar has been used to monitor forests since the 1980s and is well suited to estimate
vegetation height, volume, and biomass (Hopkinson et al., 2006; Wulder et al., 2008). Vegetation
cover percentage for canopy height above 2 m from airborne Lidar data are used to estimate the
fraction of tall versus low vegetation for LC classes with a mix of woody and herbaceous
vegetation in this study. The Lidar data were collected along 34 survey flights across the boreal
forest of Canada in the summer of 2010 by the Canadian Forest Service (Wulder et al., 2012). A
25 by 25 m tessellation was generated with the approximately 400 m wide Lidar swath, with
each cell treated as an individual Lidar plot.
A tree cover fraction (TCF) dataset for 2010 is also used in this study (Hansen et al., 2013;
hereafter the Hansen TCF dataset). It was based on Landsat images at 30 m resolution. In
contrast to the discrete LC classification datasets (providing a certain number of LC classes) as
described above, the Hansen dataset is a vegetation continuous field product (providing tree
cover fractions from 0 – 100%), in which the satellite spectral information was used to estimate
the TCF in each pixel using a regression tree algorithm (Hansen et al., 2002; 2010). This may
better represent heterogeneous areas than is possible by discrete LC classification. Tree cover is
defined to exist over pixels where canopy closure is observed for vegetation taller than 5 m in
height. Forests are generally defined as woody vegetation taller than 3 m in the regional and
global LC datasets. The different definitions of tree heights should not result in much difference
in areas with mature forests, such as most boreal forests in Canada.
Simulated surface albedo by the CLASSIC model in offline experiments is evaluated against the
Moderate Resolution Imaging Spectroradiometer (MODIS) (MCD43C3) broadband (0.3–5.0
μm) white-sky albedo (Schaaf et al., 2002), with quality flags of 0–2 (75% or more full
inversions and 25% or fewer fill values) and solar zenith angles less than 70°. The MODIS
albedo product used in this study is at 0.05 degree resolution and is regridded to the 0.22 degree
resolution used for the CLASSIC simulations (see Section 2.4.2).
**2.4 The CLASSIC model and simulation setup**
**2.4.1 The CLASSIC model**
CLASSIC is the successor to the coupled modelling framework based on the Canadian Land
Surface Scheme (CLASS; Verseghy, 1991; 1993) and the Canadian Terrestrial Ecosystem Model
(CTEM; Arora and Boer, 2005; Melton and Arora, 2016). The physics and biogeochemistry
components of CLASSIC are based on CLASS and CTEM, respectively.
For the physics component, the default model's vegetation is represented in terms of the
fractional coverage of the four PFTs (needleleaf trees, broadleaf trees, crops, and grasses). The
physics component represents a single snow layer with variable depth and a single vegetation
canopy layer. As a first-order treatment of subgrid-scale heterogeneity, each grid cell is divided
up into four sub-areas, consisting of vegetated and bare soil areas, each with and without snow
cover. The visible and near-infrared albedos of each PFT/vegetation category are specified.
These albedos are further modified by taking into account the fraction of the ground that is seen
from the sky above referred to as the sky view factor (which is modelled as a function of the leaf
area index). The albedo of the ground that is seen from the sky above depends on if the ground is
snow covered or not but also on the soil moisture of the top soil layer, since wet soil is darker
than the dry soil. Canopy snow processes such as interception/unloading, sublimation, and melt
are all simulated. The aggregated visible and near-infrared albedos for the bulk canopy are
incremented using the current values weighted by the fractional coverage of the vegetation
categories (Verseghy 1993). More details can be found in Appendix A. The overall surface
albedo of a grid cell is computed as a weighted mean using the fractional coverages for the four
sub-areas. Twenty ground layers represent the soil profile, starting with 10 layers of 0.1 m
thickness. The thicknesses of the layers gradually increase to 30 m for a total ground depth of
over 61 m. Liquid and frozen soil moisture contents, and soil temperature, are determined
prognostically for permeable soil layers.
The biogeochemistry component of CLASSIC used here represents vegetation in terms of nine
PFTs: Needleleaf Evergreen trees (NLE), Needleleaf Deciduous trees (NLD), Broadleaf
Evergreen trees (BLE), Broadleaf Cold Deciduous trees (BCD), Broadleaf Dry Deciduous trees
(BDD), $C_3$ and $C_4$ Crops (C3C/C4C), and $C_3$ and $C_4$ Grasses (C3G/C4G). These nine PFTs map
directly onto the four PFTs used by CLASSIC's physics component. When the physics and
biogeochemistry components are coupled together, as in the case of simulations carried out in
this study, the structural attributes of vegetation including leaf area index, canopy mass, rooting
depth, and vegetation height are simulated dynamically as a function of environmental
conditions by the biogeochemistry component. When the biogeochemistry component is turned
off, specified structural attributes of vegetation for use by the physics component are extracted
from look-up tables.
**2.4.2 Simulation setup**
Gridded meteorological data based on the Climatic Research Unit (CRU,
https://crudata.uea.ac.uk/cru/data/hrg/) and Japanese reanalysis (JRA) (CRUJRA) are used to
drive CLASSIC simulations. The data are constructed by regridding data from the JRA and
adjusting where possible to align with the CRU TS 4.05 data. The blended product from January,
1901 to December, 2020 has the 6-hourly temporal resolution of the reanalysis product but
monthly means adjusted to match the CRU data (Harris, 2020).  The 6-hourly data are
disaggregated on-the-fly within CLASSIC into half-hourly data following the methodology by
Melton and Arora (2016) for the following seven meteorological variables that are used to force
the model: 2 m air temperature, total precipitation, specific humidity, downward solar radiation
flux, downward longwave radiation flux, surface pressure, and wind speed. Surface temperature,
surface pressure, specific humidity, and wind speed are linearly interpolated. Long-wave
radiation is uniformly distributed across a 6-hour period, and shortwave radiation is diurnally
distributed over a day based on a grid cell's latitude and day of year with the maximum value
occurring at solar noon. Precipitation is treated following Arora (1997), where the total 6-hour
precipitation amount is used to determine the number of wet half hours in a 6-hour period. The 6-
hour precipitation amount is then spread randomly, but conservatively, over the wet half-hourly
periods. In CLASSIC, the phase of precipitation is determined by a threshold surface air
temperature with three options available (Bartlett et al., 2006). The 0°C air temperature threshold
is used to partition precipitation into rain or snow in this study. This choice does not have a
significant impact on the simulated surface albedo in CLASSIC especially during the February-
March months when the snow cover is near its maximum (Wang et al. 2014).
Two simulations over the 1850-2020 historical period are performed using PFTs derived from
the ESA-CCI and the GLC2000 datasets respectively, which is the only difference between the
two simulations. Static PFTs are used in the simulations where the fractional coverage of PFTs is
prescribed and does not vary through time. Besides land cover and meteorological forcings,
CLASSIC requires globally averaged atmospheric $CO_2$ concentration, and geographically
varying time-invariant soil texture and soil permeable depth. The atmospheric $CO_2$ concentration
values are provided by the Global Carbon Project protocol
(https://www.globalcarbonproject.org/index.htm). The soil texture information consists of the
percentage of sand, clay, and organic matter and is derived from the SoilGrids250m dataset
(Hengl et al., 2017), and permeable soil depth is based on Shangguan et al. (2017). The
simulations are performed at a 0.22 degree rotated latitude-longitude grid over a domain
including Canada and part of Alaska (Fig. 3). Pre-industrial simulations that correspond to the
year 1850 are required prior to doing the historical simulations so that model's carbon pools,
including leaf biomass which determines leaf area index, are spun up to near equilibrium for
each land cover. The pre-industrial simulations use 1901-1920 meteorological data repeatedly
with atmospheric $CO_2$ concentration specified at its 1850 level. Each historical simulation is then
initialized from its corresponding pre-industrial simulation after it has reached equilibrium (with
carbon fluxes to conditions corresponding to the year 1850). For the period 1851-1900, the
CRUJRA meteorological data for the first 20 years (1901-1920) are used repeatedly. For the
1901-2020 period the meteorological data corresponding to each actual year are used. The period
from 2001 to 2015 was selected for analyzing the simulated results.
**3. PFT mapping methods**
The CW-table for the ESA-CCI LC dataset is generated through a multi-step process that
combines multiple land cover maps at different spatial and categorical resolutions with ancillary
data on tree cover and vegetation height (Fig. 2). This includes the following steps: (1)
combining two existing land cover maps (NALCMS and VLCE) to produce a harmonized 30 m
land cover (Hybrid) map with improved categorical precision (as described in Section 2.1); (2)
creating a CW-table for the Hybrid land cover map through a direct mapping of classes from the
Hybrid map onto the CLASSIC PFTs, such that each land cover class corresponds to a particular
mix of PFTs as represented in CLASSIC. This step is supported by vegetation height data from
an airborne Lidar campaign over parts of Canada; (3) computing the sub-pixel fractional
composition for classes in the ESA-CCI land cover map (300 m resolution) based on the 30 m
Hybrid land cover dataset and the Hansen tree cover fraction dataset; (4) using the sub-pixel
fractional composition analysis to create a CW-table for mapping the ESA-CCI land cover
classes onto PFTs as represented in CLASSIC; and (5) since the ESA-CCI dataset is global, the
CW-table developed over Canada is extended to the whole globe.

## 3.1 CW-table for mapping Hybrid LC classes to CLASSIC PFTs

Among the nine CLASSIC PFTs, BLE and BDD forests are not present in Canada. These are primarily tropical PFTs as represented in CLASSIC. NLD accounts for less than 1% of coniferous forests in Canada (Wang et al., 2019). Therefore, we do not consider NLD, BLE, and BDD from here on in this study. Considering the fine resolution (30 m) of the Hybrid map, especially relative to the model resolution (~16 km) used in this study, we assign fractions of 1.0 to the two pure forest classes (LC210 and LC220), the cropland (LC15), and the five non-vegetative classes (LC16 to LC32) in its CW-table (Table 1). The mixed-wood category (LC230) is split evenly into NLE and BCD in the table based on the definition in the VLCE legend (Hermosilla et al., 2018; Wulder et al., 2003). Note that in Table 1, broadleaf deciduous trees (BDD and BCD) are considered together and separated later into their cold and drought deciduous versions. Similarly, crops and grasses ($C_3$ and $C_4$) are considered together and separated later into their $C_3$ and $C_4$ varieties. The reason for this is that the separation of broadleaf trees into their cold and deciduous phenotypes is based on latitude (Wang et al., 2006). The separation of crops and grasses based on their photosynthetic pathway ($C_3$ or $C_4$) is done based on the $C_4$ fraction from Still and Berry (2003), which is available at 1° resolution.

CLASSIC explicitly represents shrub PFTs (Meyer et al., 2021), but this work does not use that model version, and therefore the fraction of tall shrubs is assigned to one of the tree PFTs as was done in creating the CW-table for GLC2000 for use with CLASSIC (Wang et al., 2006). Four (LC2 - Sub-polar taiga needleleaf forest, LC50 - Shrubland, LC80 - Wetland, and LC81-Wetland-treed) out of the 17 classes in the Hybrid map are characterized by a mosaic of trees, shrubs, and herbaceous vegetation. The vegetation coverage for canopy height above 2 m from Lidar plots is used to inform the partitioning of forest (tall vegetation) to non-forest (low

vegetation) fractions for these mixed classes. We overlay the Lidar plots on the Hybrid land
cover map in ArcGIS. Samples (20 to 40, note that these classes do not cover large areas in
Canada) for the four mixed classes in the Hybrid map are selected where there are Lidar data.
The vegetation coverage data (for canopy height above 2 m) from Lidar plots for samples of each
class are used to compute an average coverage of tall vegetation ($> 2$ m) for that class, which is
then used to assign forest fractions for these four classes in Table 1.
The distribution of tree species from Beaudoin et al. (2014) is used to guide the separation of
coniferous versus broadleaf forest fractions. For example, for the Wetland-treed category
(LC81), maps of tree species show that coniferous forests dominate wetland-treed regions, while
broadleaf forests are generally non-existent. We, therefore, assign most of the forest fraction to
NLE (0.55), only 0.05 to BCD, 0.35 to grasses, and the remaining to the bare ground for LC81
(Table 1). The presence of evergreen shrubs is rare in Canada according to National Forest
Inventory ground plots data (Gillis et al., 2005), thus we only assign an estimated tall shrub
fraction (0.20) in the shrub class (LC50) to BCD. The sub-polar or polar classes (LC11 to LC13)
are located above the treeline and mainly consist of low shrubs and grass. The fractions of grass
(including low shrubs) and bare ground are based on field surveys of fractional vegetation cover
and tundra PFT data in Bjorkman et al. (2018) and Macander et al. (2020) (by computing the
average fractions at the field sites which overlap with the sub-polar or polar classes in the
Hybrid/NALCMS land cover map). High-resolution images from Google Earth engine or Bing
Maps are also used to examine the ratio of vegetated versus bare ground for all classes in which
bare ground is present.
**3.2. CW-table for mapping ESA-CCI LC classes to CLASSIC PFTs over Canada**

### 3.2.1 The error and sub-pixel fractional error matrices

A standard approach for the accuracy assessment of LC products is to use an error matrix. It is a
square array or table of numbers arranged in rows and columns, in which the classification from
the LC product (usually represented by the rows) is compared to the reference data (usually
represented by the columns) for each category (Congalton, 1991). The major diagonal of the
matrix presents the number of correct classifications indicating the agreement between the LC
and the reference data for each category. In practice, fine-resolution regional maps are often used
to assess large-scale LC products derived from coarse-resolution data (Cihlar et al., 2003). In
such cases, the fine-resolution reference data are aggregated/regridded to match the grid of the
coarse-resolution data. Several classes in the reference data may be present in a single coarse-
resolution pixel depending on the homogeneity of the landscape. In order to compare the
reference and the LC data on a one-to-one basis, the dominant LC class (the class with the most
abundant fractions based on all fine-resolution pixels in the reference data) is often assigned to
the regridded reference pixel.
The sub-pixel fractional error matrices have been introduced as a more appropriate way of
assessing the accuracy of mixed pixels by Latifovic and Olthof (2004). In contrast with an error
matrix where only the dominant LC class is used as described above, the sub-pixel fractional
error matrix is produced by assigning sub-dominant LC classes from all fine-resolution pixels in
the reference data to the corresponding single coarse-resolution pixel. It thus allows a
quantitative assessment of the fractional composition of the LC classes in the coarse resolution
dataset. In this study, both the 30 m Hansen TCF data and the 30 m Hybrid LC map are used to
compute the sub-pixel fractional error matrices of the 300 m ESA-CCI dataset (Table 2 and
Table 3). However, the objective here is not an accuracy assessment as in Latifovic and Olthof
(2004) but rather to obtain the fractional composition of the LC classes in the ESA-CCI product
and to inform the PFT mapping process. We refer to this process as the sub-pixel fractional
composition analysis in the rest of this paper. Sub-pixel fractional composition analyses are first
performed for each ecozone and then weighted mean fractions for each ESA-CCI class are
computed based on pixel counts in each of the ecozones (see the location of ecozones in Fig. 1).
For the Hansen TCF data, results are shown only for the ESA-CCI LC classes containing forests
in Canada (Table 2). In the ESA-CCI legend (Table 4), two sub-classes for broadleaf (LC61 and
LC62) and needleleaf (LC71 and LC72) forests are included as the closed (>40% forest cover)
and open (10-40% forest cover) categories apart from the main classes (LC60 and LC70, closed
to open (>15%)). As expected, the TCF is larger for the closed classes than for the main and the
open classes (Table 2). In Table 2, we also include ratios of TCF between the main class and the
closed class, and between the open class and the closed class. We note that the ratios are
different for broadleaf (main class vs. closed class: 68.5/86.7=0.8; open class vs. closed class:
0.43/86.7=0.43) and needleleaf (main class vs. closed class: 39.3/61.7=0.6; open class vs. closed
class: 23.2/61.7=0.38) forests, which need to be taken into account when creating the CW-table
for the ESA-CCI dataset.
To obtain representative class compositions of the ESA-CCI dataset, only homogenous ESA-CCI
pixels are included in the sub-pixel composition analyses based on the Hybrid LC data. The
homogenous pixels are defined following the method in Herold et al. (2008). To quantify
landscape heterogeneity, 3×3 pixel neighborhoods are assessed for the ESA-CCI data. A
neighborhood is considered homogenous if only one LC class is present. The weighted mean
fraction for each class is computed from ecozones with more than 10 homogenous ESA-CCI
pixels for that class (Table 3). Only 13 out of the 37 ESA-CCI classes are included in Table 3,
the rest of the classes either have limited presence in Canada or are non-vegetative (Table 4).
In the Hybrid CW-table (Table 1), four LC classes (2, 81, 210, and 230) contribute to the
fractional cover of NLE, and five LC classes (50, 80, 81, 220, and 230) contribute to the
fractional cover of BCD. In Table 3, we also include an integrated fractional cover (F) for NLE
and BCD (last two rows) for each of the ESA-CCI classes based on the following formula:

$$F = \sum_{i=1}^{N} F1_i * F2_i \qquad (1)$$

Where $F1_i$ are fractions in Table 3, $F2_i$ are fractions in Table 1, and N is the number of Hybrid
LC classes contributing to NLE (N = 4) or BCD (N = 5) as shown in Table 1. As an example, the
fraction of NLE for the LC70 (Tree cover needleleaf evergreen closed to open) in ESA-CCI data
in Table 3 (see the NLE row and the column for class 70) is obtained as follows: F = 0.02×0.20 +
0.17×0.55 + 0.29×1.0 + 0.09×0.5 = 0.44. This process reduces the subjectivity in assigning the
ESA-CCI land cover classes to CLASSIC's two tree PFTs (NLE and BCD) that are present in
Canada since the process is based on the high-resolution Hybrid LC data.
**3.2.2 CW-table for the ESA-CCI LC dataset over Canada**
Table 2 and Table 3 thus form the basis for creating the CW-table for mapping the ESA-CCI LC
classes to CLASSIC's PFTs (Fig. 2 and Table 4). For the ESA-CCI class LC61 (Tree cover
broadleaved deciduous closed, not included in Table 3 due to limited presence in Canada), ratios
of TCF for LC60 vs LC61 in Table 2 and the fractions of LC60 (Tree cover broadleaved
deciduous closed to open) in Table 3 are used to derive fractions for LC61 in Table 4. The
remapping of LC62 (Tree cover broadleaved deciduous open) and LC72 (Tree cover needleleaf
evergreen open) into CLASSIC's PFTs is done in a similar way. Since NLD is not included in
either Table 2 or Table 3, the needleleaf deciduous tree cover classes (LC80-82) are assigned to
the same fractions as the needleleaf evergreen tree cover classes (LC70-72). For simplicity, the
fractions in Table 3 are rounded to values with either "0" or "5" at the hundredth place when
used in Table 4. For the rest of the classes not included in either Table 2 or Table 3, values are
based on the default CW-table from the ESA-CCI user guide (Table 7-2, ESA, 2017). The spatial
distribution of LC classes is also taken into consideration when determining the fractions in the
CW-table. For example, the sparse vegetation class (LC150) is mainly distributed above the
treeline in alpine and Arctic tundra environments, thus we only assign 0.05 to BCD, the rest to
C3G/C4G and bare ground (Table 4).
The six CLASSIC PFTs (those present in Canada) are produced from the Hybrid and the ESA-
CCI maps based on Table 1 and Table 4 respectively. The PFTs from the Hybrid map are used as
a reference here to map ESA-CCI land cover classes to CLASSIC's PFTs. To make the spatial
distribution of PFTs from ESA-CCI agree better with those from the Hybrid dataset, fractions for
the following classes in Table 4 are adjusted upward by 0.05: LC60 from 0.65 to 0.70 for BCD;
LC71 and LC81 from 0.80 to 0.85 for NLE; and LC120 from 0.10 to 0.15 for BCD. Values for
LC10-20 are also slightly adjusted to reduce crop fractions.
Based on Table 4, the fractional coverage of nine CLASSIC PFTs are also produced on a global
scale and used in offline CLASSIC simulations in Arora et al. (2022), who carry out a
comprehensive assessment of the impact of using two different LC datasets (ESA-CCI versus
GLC2000) for representing the nine PFTs in the CLASSIC model. However, some adjustments
to Table 4 are found to be necessary. This is because fractions of NLE (Needleleaf evergreen
forests) in Eurasia are found to be too low relative to the Hansen TCF data, with maximum
values only around 0.45 in most NLE dominated areas, where the maximum TCF from the
Hansen dataset is around 0.80. This indicates that the needleleaf evergreen forests classes (LC
70-72) in the ESA-CCI map may represent different forest/tree cover fractions in Canada and
Eurasia, which is confirmed by sub-pixel fractional composition analyses based on the Hansen
TCF dataset. Details are presented in Appendix B.

**4. Results**

**4.1 Comparison of PFTs from Hybrid, ESA-CCI, and GLC2000 data**

Figure 3 shows the spatial distribution of PFTs derived from the Hybrid, ESA-CCI, and
GLC2000 LC datasets respectively. $C_4$ crops (C4C) and grasses (C4G) are sparse in Canada as
would be expected since C4 PFTs grow only in warmer temperatures when the average monthly
temperature exceeds 22 °C (Fox et al., 2018). Based on the fractional distribution of C4
vegetation in Still and Berry (2003) and the Hybrid map, the average fraction is 0.5% for C4
crops and 0.1% for $C_4$ grasses in Canada. Therefore, only four out of the six PFTs (those present
in Canada) are shown in Figure 3. The $C_4$ fraction product from Still and Berry (2003) is available
at a much coarser spatial resolution (1°) than other land cover products used in this study, and it is
a global product. As such then, the estimated $C_4$ fractions for crops/grasses in Canada used here
may not completely agree with those from regional estimates. In general, the spatial distributions
of the PFTs from the ESA-CCI and the Hybrid datasets agree well except for $C_3$ grasses (C3G)
(Fig. 3j and Fig. 3k). This is not surprising given that the CW-table for the ESA-CCI dataset is
based on the Hybrid map. Areas mapped as C3G in Hybrid (Fig. 3j), were mainly classified as
sparse vegetation (LC150) in the ESA-CCI legend (Table 4). However, LC150 from ESA-CCI
was also found in some areas of the high Arctic islands, where barren land is the dominant class
in the Hybrid map (grey coloured areas in Fig. 1). If too much grass were assigned to LC150, it
would yield unrealistically large fractional coverage of grass in the high Arctic islands. In Table
4, for LC150, 0.05 is assigned to BCD, 0.35 to grasses, and the rest to the bare ground for
LC150, which yields a total vegetation cover of 40% and is more than the definition (<15%
vegetation) used in the ESA-CCI legend. Yet, this still results in less C3G and less bare ground
in the ESA-CCI map (Fig. 3k and Fig. 3n) than those from the Hybrid map (Fig. 3j and Fig. 3m).
This suggests that it is not ideal to classify areas in the high Arctic islands and in the Arctic
tundra region as being in the same land cover category.
There are large differences in the spatial distribution of the PFTs based on the GLC2000 LC
product and those based on the Hybrid and ESA-CCI datasets (Fig. 3 and Fig. 4). Relative to
PFTs from ESA-CCI, GLC2000 has less NLE and more BCD in northwestern Canada, and more
NLE in southern and eastern Canada (Fig. 4a and Fig. 4b). GLC2000 based CLASSIC PFT
fractions also exhibit more crops, less grass, and more bare ground (Fig. 4c-4e). These
differences partly stem from the differences in the ESA-CCI and GLC2000 LC datasets, but are
also due to the fact how the fractions in the CW-tables of the two datasets are used to translate
LC data to fractional coverage of PFTs as demonstrated in Wang et al. (2019).
**4.2 Bias in simulated surface albedo and LAI**
The top row of Figure 5 shows the bias in winter albedo (March) simulated by CLASSIC when
using PFT distributions based on the ESA-CCI (Fig. 5a) and GLC2000 products (Fig. 5b). While
model biases are the result of both the driving geophysical and meteorological data that are used
to force the model, as well as the model itself, the comparison between the two simulations does
show the effect of differences in the distribution of PFTs.  Relative to observed surface albedo
from MODIS, there are relatively large negative biases in the southwest of Hudson Bay and
central Quebec, while there are relatively large positive biases in western Canada and Alaska in
the simulation when using the GLC2000 product to obtain PFT distributions (Fig. 5b). Both the
negative and the positive biases are largely reduced in the simulation using PFT distributions
based on the ESA-CCI product (Fig. 5a). The lower row of Figure 5 shows the spatial
distribution of the difference in surface albedo (Fig. 5c) and leaf area index (Fig. 5d) between the
two simulations, which are closely correlated (r = -0.85). Given the same meteorological forcing
dataset is used to drive both simulations, the differences in the simulated LAI are due mainly to
the different PFT distributions used in the two simulations. Since NLE is the only PFT with
LAI > 0 during winter in Canada, the LAI difference in March as shown in Figure 5d is mainly
due to the different fractional coverage of NLE based on the ESA-CCI and GLC2000 products
(Fig. 4a).
In contrast, the large positive albedo biases (up to ~ 0.4) in southern Canada are more or less the
same in both simulations (Fig. 5a and Fig. 5b), where the dominant PFT is C3 crops (Fig. 3h and
Fig. 3i). Those positive albedo biases are likely due to the standing crop stubble and the lack of
the representation of blowing snow and its sublimation currently in CLASSIC (Harder et al.,
2018; Pomeroy et al., 1993). Harder et al. (2018) showed that the height of the stubble over
wheat and canola field in Saskatchewan, Canada may range from 10 to 40 cm, with a maximum
PAI (plant area index) of 1.0.  Wang et al. (2016) showed that surface albedo in CLASSIC
decreased exponentially with increasing PAI for the bare or snow-covered canopy over snow,
while most reductions of the albedo were achieved through the increase of PAI from 0 to 1.0.
They showed that surface albedo decreased from 0.75 to 0.31 in CLASSIC when PAI increased
from 0 to 1.0 for the bare canopy over snow, which appears to account for most of the positive
albedo biases in the agricultural areas of southern Canada (Fig. 5a and Fig. 5b). Improvements to
the crop module of CLASSIC to improve cropland albedo are currently being considered.
**5. Summary and conclusions**
A hybrid land cover map at 30 m resolution is created by merging the NALCMS and VLCE land
cover products over Canada. Vegetation height data from Lidar plots, tree species, and high
resolution images are used to inform the creation of a CW-table for mapping the 17 LC classes
of the Hybrid map to six CLASSIC PFTs that are present in Canada. Both the Hybrid map and
the Hansen tree cover fraction data are used to compute the sub-pixel fractional composition of
the LC classes in the ESA-CCI LC dataset, which is then used to create a cross-walking table for
mapping the 37 ESA-CCI categories to CLASSIC PFTs over the model's Canadian domain.
Based on the new CW-tables, PFT distributions are produced from the Hybrid and the ESA-CCI
LC products, respectively, and are compared with those based on the GLC2000 dataset currently
used in CLASSIC. The results show that the spatial distribution of PFTs from the ESA-CCI
dataset is in better agreement with those from the Hybrid map, while there are large differences
in the PFTs from the GLC2000 dataset and from the Hybrid/ESA-CCI datasets. The CW-table
developed over Canada is adjusted and used to map PFTs based on the ESA-CCI LC product for
use in CLASSIC simulations at the global scale.
Our PFT mapping approach for the ESA-CCI dataset is mainly based on sub-pixel fractional
composition analyses using the Hybrid map and the Hansen tree cover fraction data, and
therefore the accuracy of the latter two datasets affects the PFT mapping process. Some LC
categories in the ESA-CCI legend either have limited presence or no presence in Canada, such as
the Needleleaf deciduous trees, Broadleaf Evergreen trees, and Broadleaf Dry Deciduous trees
etc., and the sub-pixel fractional composition analyses therefore can not be performed for these
LC categories. The needleleaf deciduous tree cover classes are assigned to the same fractions as
the needleleaf evergreen tree cover classes in the CW-table, and values based on the default CW-
table from the ESA-CCI user guide are used for the other LC categories. Therefore, potentially
large uncertainties may be associated with these classes in the resulting fractional coverage of
PFTs especially at the global scale. Similar analyses for other regions (e.g. Eurasia and tropics)
for which high quality regional land cover maps are available will be helpful in reducing these
uncertainties in the future work. In addition, the exercise of mapping PFTs at the global scale in
this study reveals that there are inconsistencies in the representation of fractional coverage for
some LC categories in the ESA-CCI map for different regions of the globe. Future improvements
in the consistency of the LC categories globally in the ESA-CCI LC product would greatly
benefit the land surface and the earth system modelling community. In the meantime, caution
should be exercised when using this product for mapping PFTs represented in any LSM based on
a single cross-walking table at the global scale.
CLASSIC simulations driven with meteorological data from the CRU-JRA product show that the
simulated winter albedo is improved when using PFT distributions based on the ESA-CCI LC
product compared to that based on the GLC2000 product, which is consistent with findings from
previous studies. While CLASSIC simulations could also have been performed using its PFT
distributions based on the Hybrid LC product, the reason for using the ESA-CCI based PFT
fractions for CLASSIC is that ESA-CCI is a global product. CLASSIC simulations are routinely
performed at the global scale both in the framework of the Canadian Earth System Model (Swart
et al., 2019), where CLASSIC serves as its land component, and offline where global CLASSIC
simulations driven with the CRU-JRA meteorological data contribute to the annual global carbon
budget assessments of the Global Carbon Project (Friedlingstein et al., 2020; Seiler et al., 2021).
Untreated crop stubble appears to be contributing to the positive winter albedo biases in southern
Canada, which needs to be addressed in a future version of CLASSIC. These results underscore
the importance of accurate representation of vegetation distribution in a realistic simulation of
surface albedo in LSMs.
Previous methods for mapping PFTs from LC datasets have mainly been based on class
descriptions, expert knowledge, and the spatial distribution of global biomes, which is a largely
subjective process. As a consequence, a PFT method developed for mapping one LC dataset to
PFTs represented in one model can not be easily transferred to other LC datasets even for
deriving PFTs in the same model. The development of satellite and computing technology has
enabled the creation of more detailed global LC products at finer spatial resolutions in recent
years, however, the lack of an objective PFT mapping method impedes the implementation of the
new improved LC products in LSMs. Here, we have proposed a method to inform the cross-
walking process using sub-pixel fractional composition analyses based on a tree cover fraction
dataset and a fine-resolution LC map. Our results suggest that the sub-pixel fractional
composition analyses provide an effective way to reduce uncertainties in the cross-walking
process and therefore, to some extent, objectifies the otherwise subjective process. The PFT
mapping approach developed in this study can also be applied to other LC datasets for mapping
PFTs used in other LSMs.

**Appendix A**
In CLASSIC, the surface albedo for a canopy over snow ($\alpha$) is:
$$\alpha = \alpha_c(1-\chi)(1-f_{snow}) + \alpha_{c,snow}(1-\chi)(f_{snow}) + \alpha_{snow}\chi\tau_c \qquad (1)$$

$$\chi = \exp(-K*PAI) \qquad (2)$$


calculated using separate parameters ($\alpha_c$, $\alpha_{c,snow}$, $\tau_c$ and $K$) for both the visible (VIS) and near
infrared (NIR) bands, where $\alpha_c$ is the snow-free canopy albedo, $\alpha_{c,snow}$ the snow-covered canopy
albedo, $f_{snow}$ the fraction of the canopy with snow on it, $\alpha_{snow}$ the snowpack albedo. $\tau_c$ is canopy
transmissivity and is modeled using a Beer's law approach, ignoring multiple reflections
(Verseghy *et al.* 1993). *K* is an extinction coefficient that varies with vegetation type. The
appearance of $\tau_c$ in the last term of Eq.1 accounts for the shading of the snowpack by the canopy,
converting the simulated snowpack albedo to an effective value of the canopy gaps. *PAI* is plant
area index which is the sum of leaf area index and stem area index.

**Appendix B**
Based on Table 4, the fractional coverage of nine CLASSIC PFTs are also produced on a global
scale. However, some adjustments to Table 4 were found necessary. This is because fractions of
NLE (Needleleaf evergreen forests) in Eurasia are found to be too low relative to the Hansen
TCF data, with maximum values of only around 0.45 in most NLE dominated areas, where the
maximum TCF from the Hansen dataset is around 0.80. Needleleaf evergreen forests are
represented by LC classes 70 (closed to open), 71 (closed), and 72 (open). Examining the ESA-
CCI LC map shows that in Eurasia nearly all needleleaf evergreen forests are classified as LC70
(closed to open), with only less than 400 pixels as LC71 (closed), and none as LC72 (open). In
contrast, in Canada 36% of needleleaf evergreen forest are classified as LC70 (closed to open),
64% as LC71 (closed), and less than 1% as LC72 (open). This is understandable given that sub-
classes were only assigned where surface samples were available (ESA, 2017). Sub-pixel
fractional composition analyses of the ESA-CCI classes based on the Hansen TCF dataset show
that in Eurasia TCF for LC70 (closed to open) is 66% and for LC71 (closed) is 35% (note the
few pixels within this class). This is in contrast with those in Canada where the TCF for LC70
(closed to open) is 39% and for LC71 (closed) is 62%, explaining the too low NLE fractions in
Eurasia when mapping PFTs based on Table 4, and also the too high TCF in northwestern
Canada when mapping PFTs based on the default CW-table (Wang et al., 2018). In order to
apply Table 4 globally, the original LC70 (closed to open) was split into two classes: LC73 (a
new class) which is the same as LC70 over Canada (and zero everywhere else), and LC70
(revised) which is the same as before except zero over Canada. The fractions for the new LC70
class are made the same as for LC71 in Table 4, which applies to NLE outside of Canada.
Essentially, the closed-to-open needleleaf forest LC70 class over Eurasia is treated as the closed
needleleaf forest.
**Code and data availability**
More information about the CLASSIC land surface model and its Fortran code are available at
https: //cccma.gitlab.io/classic_pages/ (Melton, 2022). The VLCE LC map and the Lidar plots
data were from the Canadian Forest Service of Natural Resources Canada
(https://opendata.nfis.org/mapserver/nfis-change_eng.html). The NALCMS LC map was from
the Canada Centre for Remote Sensing (https://open.canada.ca/data/en/dataset/c688b87f-e85f-
4842-b0e1-a8f79ebf1133). The MODIS albedo data were from the online data pool at the NASA
Land Processes Distributed Active Archive Center (LP DAAC;
https://lpdaac.usgs.gov/data_access). The tree cover fraction data were from the Global Land
Analysis and Discovery laboratory in the Department of Geographical Sciences at the University
of Maryland (https://glad.umd.edu/Potapov/TCC_2010/). The GLC2000 LC map was
downloaded from the EU Science Lab
(https://forobs.jrc.ec.europa.eu/products/glc2000/data_access.php). The ESA-CCI LC map was
provided by ESA (https://www.esa-landcover-cci.org).

**Author contributions**

LW conceived and designed this research and wrote the majority of the manuscript. VKA
contributed to the writing on the CLASSIC model description and simulation setup. PB
contributed to discussions about the research. EC designed the reprojection/regridding method
for all the satellite-derived datasets and computed the PFT data. SRC processed field surveys of
vegetation cover data and helped with model simulations. VKA, EC, and SRC provided
comments and suggestions on the entire paper.

**Competing interests**

The contact author has declared that none of the authors has any competing interests.

**Acknowledgements**

We'd like to thank Mike Brady (ECCC) for his help with data processing, Joe Melton (ECCC)
for helpful comments to an early draft of the manuscript, and Mike Wulder from the Canadian
Forest Service of Natural Resources Canada for providing the Lidar plots data and for helping
interpret the Lidar data and the VLCE land cover map. We also thank Benjamin Bond Lamberty
for taking this paper on as an associate editor and the two anonymous reviewers for providing
helpful comments which greatly improved this paper.

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

Table 1. Cross-walking table for mapping the 30 m Hybrid land cover map to CLASSIC PFTs in Canada.
Nine PFTs in CLASSIC: NLE - Needleleaf Evergreen trees, NLD - Needleleaf Deciduous trees, BLE -
Broadleaf Evergreen trees, BCD - Broadleaf Cold Deciduous trees, BDD - Broadleaf Dry Deciduous
trees, C3C - C3 Crops, C4C - C4 Crops, C3G - C3 Grasses, and C4C - C4 Grasses.

| ID | Map description | 1 NLE | 2 NLD | 3 BLE | 4+5 BCD BDD | 6+7 C3C C4C | 8+9 C3G C4G | Urban | Lake | Bare |
|---|---|---|---|---|---|---|---|---|---|---|
| 2 | Sub-polar taiga needleleaf forest | 0.20 | | | | | 0.60 | | | 0.20 |
| 11 | Sub-polar or polar shrubland-lichen-moss | | | | | | 0.65 | | | 0.35 |
| 12 | Sub-polar or polar grassland-lichen-moss | | | | | | 0.45 | | | 0.55 |
| 13 | Sub-polar or polar barren-lichen-moss | | | | | | 0.10 | | | 0.90 |
| 15 | Cropland | | | | | 1.0 | | | | |
| 16 | Barren lands | | | | | | | | | 1.0 |
| 17 | Urban | | | | | | | 1.0 | | |
| 20 | Water | | | | | | | | 1.0 | |
| 31 | Snow_ice | | | | | | | | | 1.0 |
| 32 | Rock_rubble | | | | | | | | | 1.0 |
| 50 | Shrubland | | | | 0.20 | | 0.60 | | | 0.20 |
| 80 | Wetland | | | | 0.05 | | 0.85 | | | 0.10 |
| 81 | Wetland-treed | 0.55 | | | 0.05 | | 0.35 | | | 0.05 |
| 100 | Herbs | | | | | | 0.80 | | | 0.20 |
| 210 | Coniferous | 1.0 | | | | | | | | |
| 220 | Broadleaf | | | | 1.0 | | | | | |
| 230 | Mixedwood | 0.50 | | | 0.50 | | | | | |

**Table 2**. The sub-pixel fractional tree cover fraction for ESA-CCI (European Space Agency - Climate
Change Initiative) land cover classes (with forest cover) based on the Hansen TCF (Tree Cover Fraction)
dataset in Canada. Ratios of TCF between the main class and the closed class, and between the open class
and the closed class are also included.

| ESA-CCI class | ESA-CCI class description | Tree cover Fraction (%) | Ratio of TCF relative to closed class |
|---|---|---|---|
| 30 | Mosaic cropland (>50%) / natural vegetation (<50%) | 13.7 | |
| 40 | Mosaic natural vegetation (>50%) / cropland (<50%) | 45 | |
| 60 | Tree cover broadleaved deciduous closed to open (>15%) | 68.5 | 0.8 |
| 61 | Tree cover broadleaved deciduous closed (>40%) | 86.7 | 1 |
| 62 | Tree cover broadleaved deciduous open (15-40%) | 37.4 | 0.43 |
| 70 | Tree cover needleleaf evergreen closed to open (>15%) | 39.3 | 0.6 |
| 71 | Tree cover needleleaf evergreen, closed (>40%) | 61.7 | 1 |
| 72 | Tree cover needleleaf evergreen open (15-40%) | 23.2 | 0.38 |
| 90 | Tree cover Mixed | 80.9 | |
| 100 | Mosaic tree and shrub (>50%) / herbaceous cover (<50%) | 37.3 | |
| 110 | Mosaic herbaceous cover (>50%) / tree and shrub (<50%) | 19.6 | |
| 120 | Shrubland | 28.1 | |
| 150 | Sparse vegetation (tree shrub herbaceous cover) (< 15%) | 4 | |
| 160 | Tree cover, flooded fresh/brackish | 43 | |
| 180 | Shrub or herbaceous cover, flooded | 26.9 | |









Table 3. The sub-pixel fractional composition for ESA-CCI classes (columns, homogenous ESA-CCI pixels) based on the Hybrid land cover map (rows) for dominant land cover classes in Canada. The fractions for NLE and BCD are computed based on equation (1).

| Hybrid/ ESACCI Class | Hybrid description | 30 | 40 | 60 | 70 | 71 | 90 | 100 | 120 | 130 | 140 | 150 | 160 | 180 |
|---|---|---|---|---|---|---|---|---|---|---|---|---|---|---|
| 2 | Sub-polar taiga needleleaf forest | | | | 0.02 | | | 0.01 | | 0.01 | | | | |
| 11 | Sub-polar/polar shrubland-lichen-moss | | | | | | | | | | 0.01 | 0.05 | | |
| 12 | Sub-polar/polar grassland-lichen-moss | | | | 0.04 | | | | 0.03 | 0.01 | 0.24 | 0.27 | 0.03 | 0.04 |
| 13 | Sub-polar/polar barren-lichen-moss | | | | 0.02 | | | 0.01 | 0.02 | 0.01 | 0.34 | 0.09 | | 0.02 |
| 15 | Cropland | 0.92 | 0.37 | 0.02 | | | | | | 0.1 | | | | |
| 16 | Barren lands | | | | | | | | | 0.01 | 0.15 | 0.17 | | |
| 50 | Shrubland | 0.01 | 0.07 | 0.06 | 0.13 | 0.05 | 0.04 | 0.32 | 0.46 | 0.09 | 0.14 | 0.25 | 0.06 | |
| 80 | Wetland | | 0.03 | 0.08 | 0.2 | 0.05 | 0.03 | 0.27 | 0.2 | 0.02 | 0.06 | 0.09 | 0.37 | 0.75 |
| 81 | Wetland treed | | 0.01 | 0.01 | 0.17 | 0.07 | 0.03 | 0.11 | 0.12 | | | | 0.43 | 0.15 |
| 100 | Herbs | 0.06 | 0.27 | 0.08 | 0.02 | | 0.02 | 0.06 | 0.09 | 0.72 | 0.01 | 0.03 | 0.01 | 0.01 |
| 210 | Coniferous | | 0.01 | 0.02 | 0.29 | 0.72 | 0.07 | 0.04 | 0.03 | | 0.01 | 0.02 | 0.06 | |
| 220 | Broadleaf | 0.01 | 0.13 | 0.57 | 0.02 | 0.01 | 0.28 | 0.07 | 0.01 | 0.01 | | | 0.01 | |
| 230 | Mixedwood | | 0.1 | 0.14 | 0.09 | 0.07 | 0.52 | 0.12 | 0.03 | | | | 0.02 | |
| NLE | Needleleaf evergreen | | 0.07 | 0.09 | 0.44 | 0.8 | 0.32 | 0.19 | 0.16 | 0.01 | 0.02 | 0.05 | 0.31 | 0.08 |
| BCD | Broadleaf cold deciduous | 0.01 | 0.19 | 0.66 | 0.09 | 0.06 | 0.57 | 0.18 | 0.09 | 0.02 | 0.02 | 0.03 | 0.05 | 0.03 |

Table 4. Cross-walking table for mapping ESA-CCI land cover dataset to CLASSIC PFTs.

| ID | ESA-CCI class description | 1 NLE | 2 NLD | 3 BLE | 4+5 BCD BDD | 6+7 C3C C4C | 8+9 C3G C4G | Urban | Lake | Ocean | Bare |
|---|---|---|---|---|---|---|---|---|---|---|---|
| 10 | Cropland, rainfed (CR) | | | | | 0.80 | 0.20 | | | | |
| 11 | CR Herbaceous cover | | | | | 0.90 | 0.10 | | | | |
| 12 | CR Tree or shrub cover | | | | 0.60 | | 0.30 | | | | 0.10 |
| 20 | Cropland, irrigated or post-flood | | | | 0.05 | 0.85 | 0.10 | | | | |
| 30 | Mosaic cropland (>50%) / natural vegetation (tree, shrub, herb) | 0.05 | | | 0.15 | 0.60 | 0.20 | | | | |
| 40 | Mosaic natural vegetation (tree,shrub, herb) >50% / crop | 0.10 | | | 0.20 | 0.40 | 0.30 | | | | |
| 50 | Tree cover broadleaved evergreen closed to open | | | 0.95 | 0.05 | | 0.0 | | | | |
| 60 | Tree cover broadleaved deciduous closed to open | | | | 0.70 | | 0.25 | | | | 0.05 |
| 61 | Tree cover broadleaved deciduous closed | | | | 0.90 | | 0.10 | | | | |
| 62 | Tree cover broadleaved deciduous open | | | | 0.40 | | 0.40 | | | | 0.20 |
| 70 | Tree cover needleleaf evergreen closed to open | 0.85 | | | 0.05 | | 0.10 | | | | |
| 71 | Tree cover needleleaf evergreen, closed | 0.85 | | | 0.05 | | 0.10 | | | | |
| 72 | Tree cover needleleaf evergreen open | 0.35 | | | 0.10 | | 0.40 | | | | 0.15 |
| 73 | Replace LC70 in Canada | 0.45 | | | 0.10 | | 0.30 | | | | 0.15 |
| 80 | Tree cover needleleaf deciduous closed to open | 0.05 | 0.40 | | 0.10 | | 0.35 | | | | 0.10 |
| 81 | Tree cover needleleaf deciduous closed | 0.05 | 0.80 | | 0.05 | | 0.15 | | | | |
| 82 | Tree cover needleleaf deciduous open | 0.05 | 0.30 | | 0.10 | | 0.45 | | | | 0.15 |
| 90 | Tree cover Mixed | 0.25 | 0.05 | | 0.60 | | 0.10 | | | | |
| 100 | Mosaic tree and shrub (>50%) / herbaceous cover (<50%) | 0.15 | 0.05 | | 0.20 | | 0.45 | | | | 0.15 |
| 110 | Mosaic herbaceous cover (>50%) / tree and shrub (<50%) | 0.05 | 0.05 | | 0.10 | | 0.65 | | | | 0.15 |
| 120 | Shrubland | | | | 0.30 | | 0.45 | | | | 0.25 |
| 121 | Shrubland evergreen | 0.15 | | 0.15 | | | 0.45 | | | | 0.25 |
| 122 | Shrubland deciduous | | | | 0.30 | | 0.45 | | | | 0.25 |

| 130 | Grassland | | | | | | 0.70 | | | | | 0.30 |
|---|---|---|---|---|---|---|---|---|---|---|---|---|
| 140 | Lichens and mosses | | | | | | 0.20 | | | | | 0.80 |
| 150 | Sparse vegetation (tree shrub herbaceous cover) (< 15%) | | | | 0.05 | | 0.35 | | | | | 0.60 |
| 151 | Sparse tree (<15%) | | | | 0.05 | | 0.35 | | | | | 0.60 |
| 152 | Sparse shrub (<15%) | | | | | | 0.30 | | | | | 0.70 |
| 153 | Sparse herbaceous cover (<15%) | | | | | | 0.30 | | | | | 0.70 |
| 160 | Tree cover, flooded fresh/brackish | 0.30 | | | 0.10 | | 0.45 | | | 0.1 | | 0.05 |
| 170 | Tree cover, flooded saline water | 0.30 | | | 0.10 | | 0.40 | | | | 0.1 | 0.10 |
| 180 | Shrub or herbaceous cover, flooded | 0.10 | | | 0.05 | | 0.45 | | | 0.15 | 0.15 | 0.10 |
| 190 | Urban areas | 0.025 | | | 0.025 | | 0.15 | 0.75 | 0.05 | | | |
| 200 | Bare areas | | | | | | | | | | | 1.0 |
| 201 | Consolidated bare areas | | | | | | | | | | | 1.0 |
| 202 | Unconsolidated bare areas | | | | | | | | | | | 1.0 |
| 210 | Water bodies | | | | | | | | 1.0 | | | |
| 220 | Permanent snow and ice | | | | | | | | | | | 1.0 |

















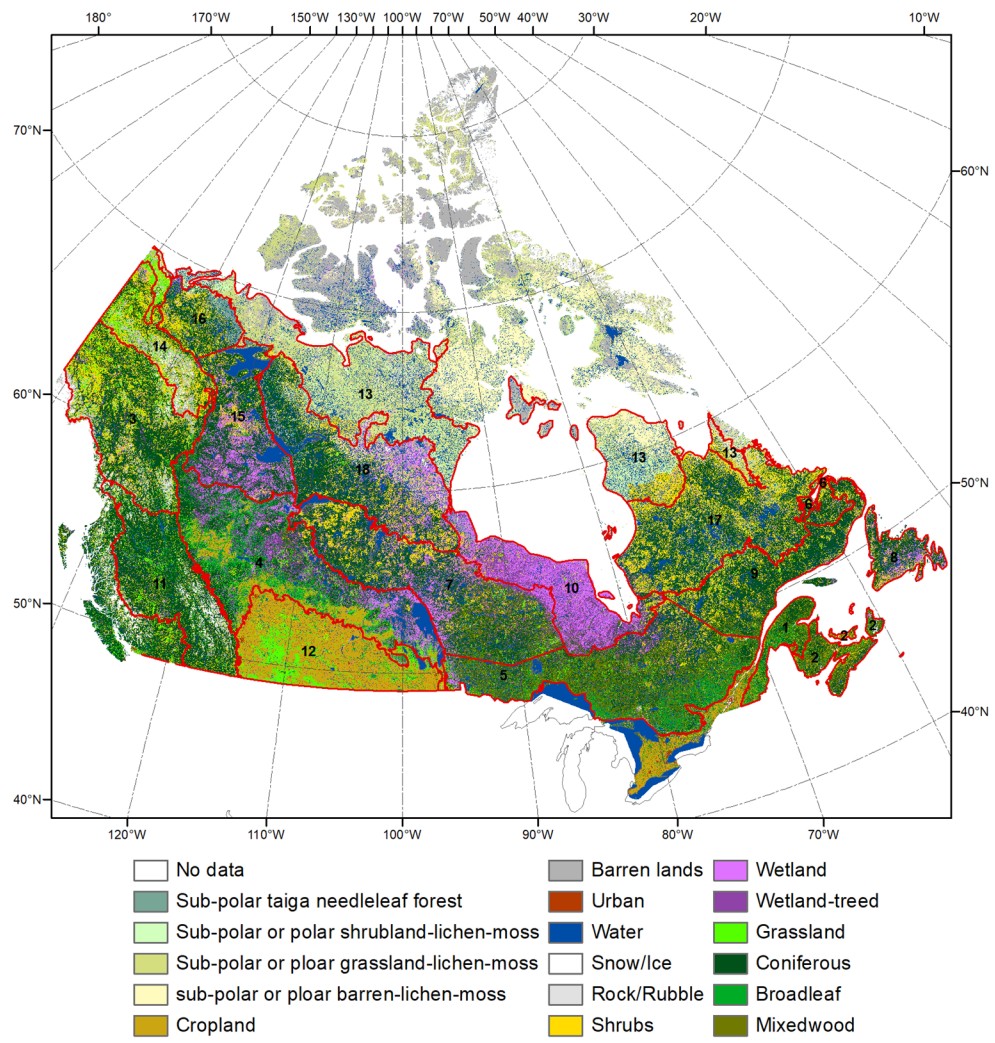


**Figure 1**. The Hybrid land cover map of Canada based on VLCE and NALCMS land cover maps for
2010. The red polygons represent 18 ecozones used in this study.







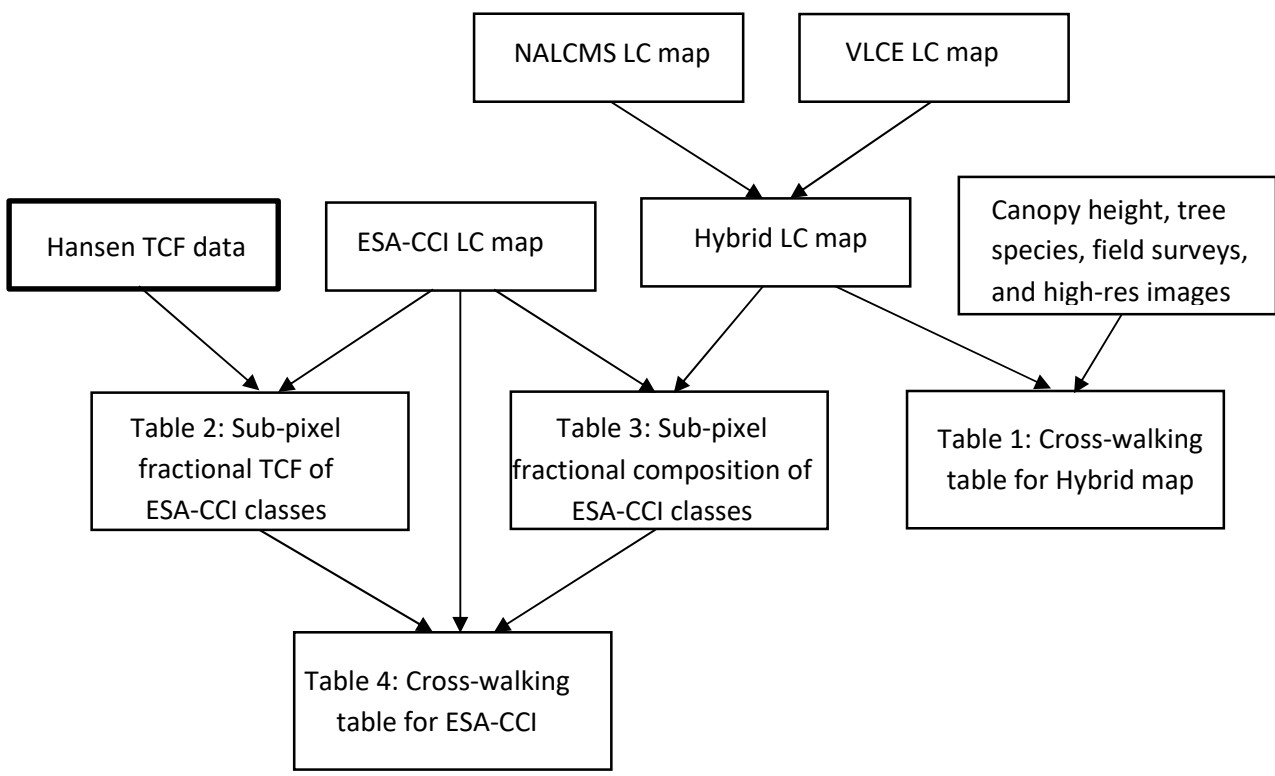

**Figure 2**. Schematic flow chart of the process for creating the cross-walking table for ESA-CCI land
cover (LC) dataset. NALCMS: the North America Land Change Monitoring System; VLCE: the Virtual
Land Cover Engine; TCF: Tree Cover Fraction.












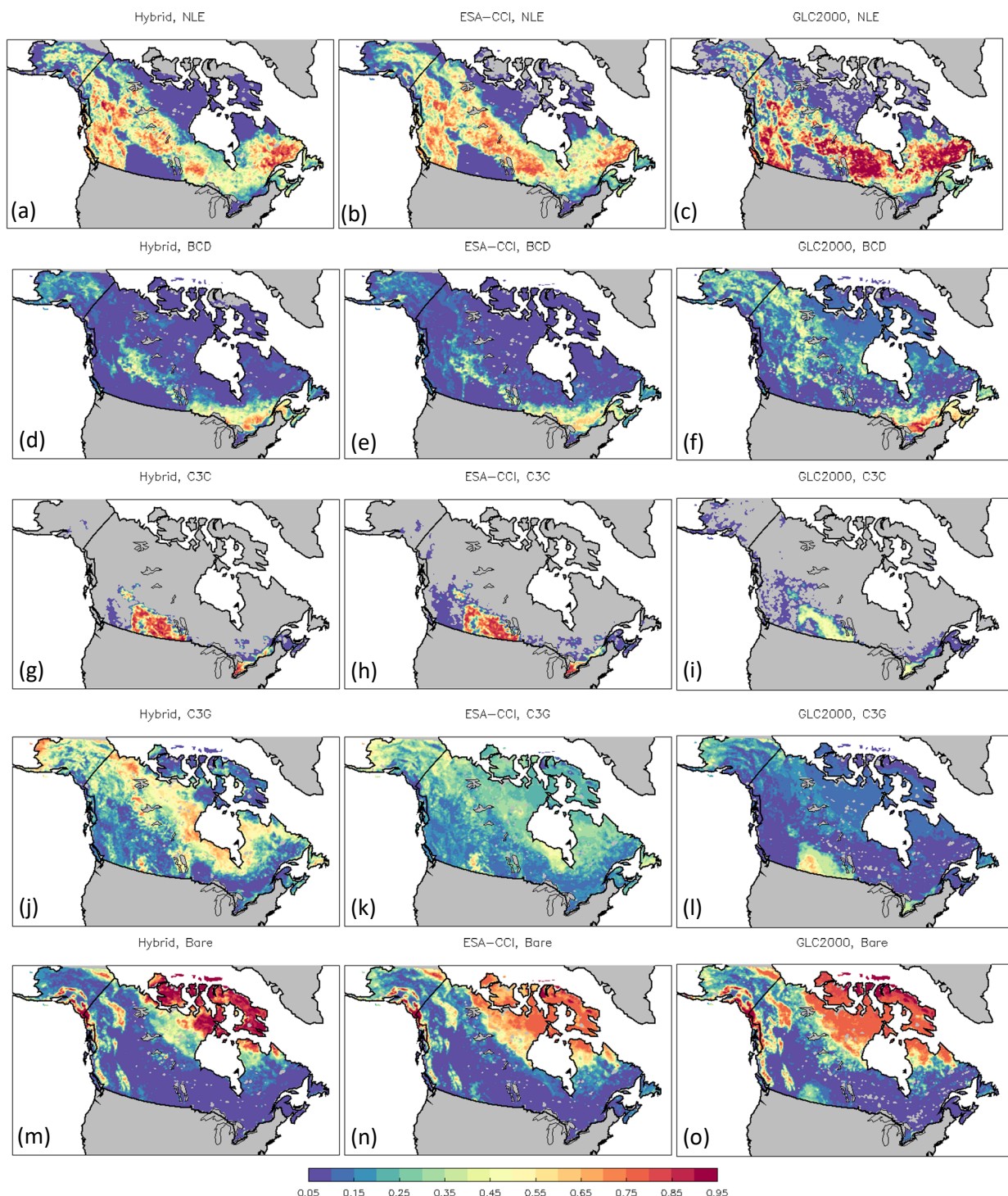


**Figure 3**. The spatial distribution of CLASSIC PFTs based on the Hybrid (left), ESA-CCI (middle), and
GLC2000 (right) land cover datasets respectively. The maps for C4C and C4G are not shown for their
fractions are small (0.5% for C4 crops and 0.1% for C4 grasses) in Canada. The last panel shows fractions
for bare ground from the three datasets.

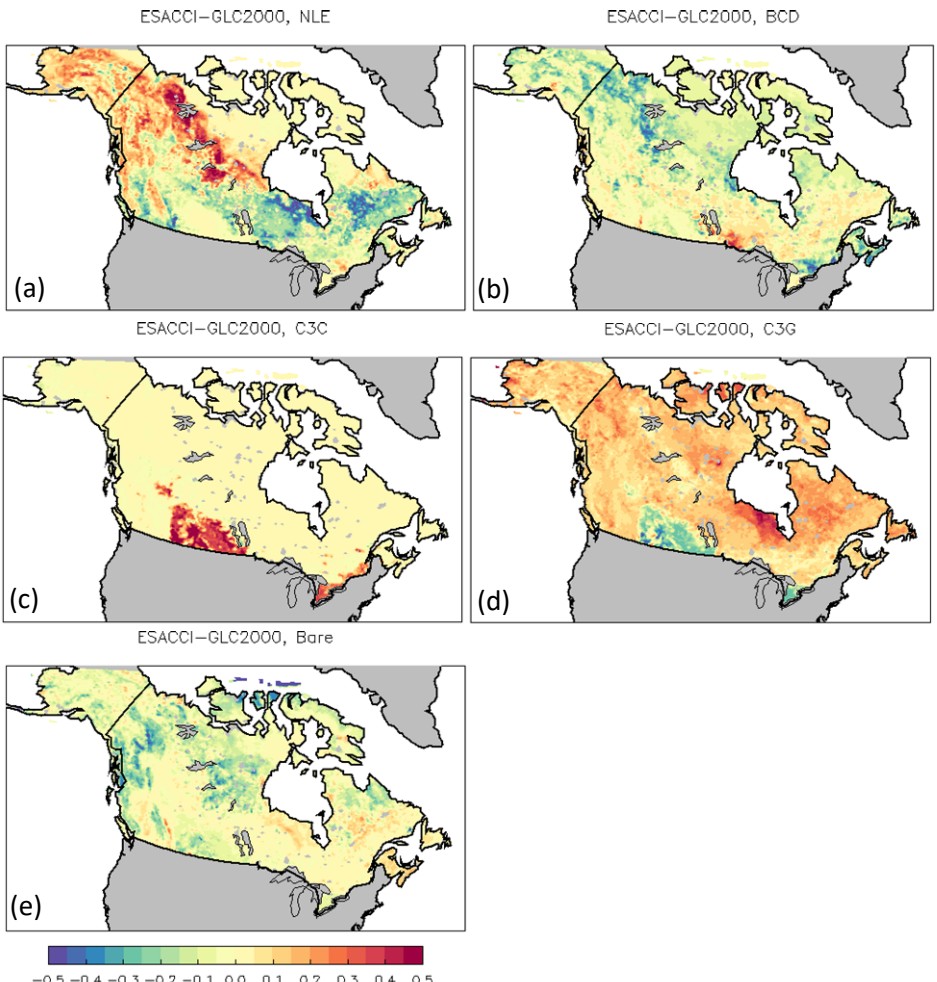

**Figure 4**. The difference in PFTs based on ESA-CCI and GLC2000 datasets for selected PFTs (a) NLE, (b) BCD, (c) C3C, (d) C3G, and (e) Bare.











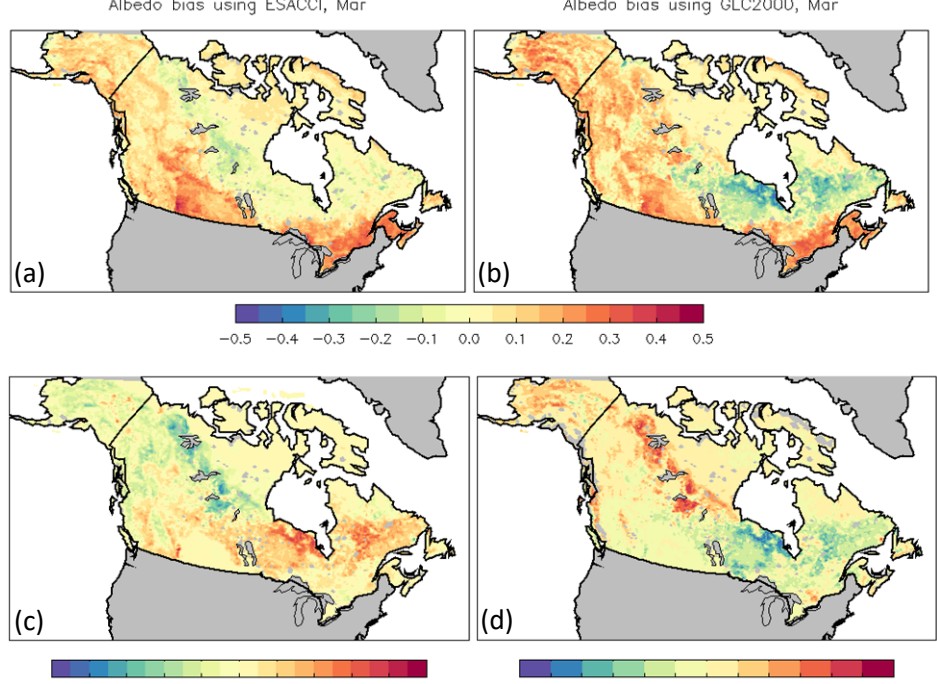

1000

Figure 5. Surface albedo bias (relative to MODIS) in CLASSIC simulations using PFT distributions
based on (a) ESA-CCI, and (b) GLC2000 land cover products. Panels (c) and (d) show the difference in
simulated surface albedo (c) and leaf area index (d) between the two simulations.

1004