# Peer review of "Mapping of ESA-CCI land cover data to plant functional types for use in the CLASSIC land model"

_EGUsphere, 2022_

## Author Comment (AC1)

EGUsphere, referee comment RC1
https://doi.org/10.5194/egusphere-2022-923-RC1, 2022
**Comment on egusphere-2022-923**

Anonymous Referee #1

Referee comment on "Mapping of ESA-CCI land cover data to plant functional types for use in the CLASSIC land model" by Libo Wang et al., EGUsphere, https://doi.org/10.5194/egusphere-2022-923-RC1, 2022

**We thank Referee #1 for their helpful comments. Our replies to his/her comments are shown in bold below.**

General comments:

This study evaluates the impact of uncertainties and biases in plant functional type (PFT) maps that are used as inputs to land surface models. The specific aim is to quantify the impact of a revised PFT map on winter albedo simulations by the Canadian Land Surface Scheme Including Biogeochemical Cycles (CLASSIC) land surface model. The improved PFT map is generated through a multi-step process that combines multiple land cover maps at different spatial and categorical resolutions with ancillary data on tree cover and vegetation height. First, the authors combine two existing land cover maps (North America Land Change Monitoring System, NALCMS; and Virtual Land Cover Engine, VLCE) to produce a harmonized 30 m land cover map for North America with improved categorical precision (i.e., more precise and accurate category labels). Next, the authors perform a direct mapping of classes from this hybrid land cover map onto the CLASSIC PFT scheme, such that each land cover class corresponds to a particular mix of PFTs as represented in CLASSIC. This step is supported by insights from vegetation height data from an airborne LiDAR campaign over parts of Canada. Next, the authors jointly combine the 30 m hybrid land cover dataset above with a 30 m tree cover fraction dataset (based on the Hansen Landsat analysis) to map sub-pixel fractional composition for classes in the European Space Agency (ESA) Climate Change Initiative (CCI) land cover map (300 m spatial resolution). Next, the authors use this analysis to map the ESA-CCI land cover classes onto PFT mixtures as represented in CLASSIC. Since the ESA-CCI dataset is global, this then allows the authors to perform CLASSIC simulations globally (with some corrections based on exploratory analysis of the resulting PFT map). Finally, the authors perform simulations for Canada and Alaska with the CLASSIC model using its original PFT map (GLC2000) and the revised ESA-CCI scheme described above, specifically looking at differences in simulated winter albedo (which is also compared to the MODIS MCD43C3 white-sky albedo product). Results show that albedo predictions are generally more accurate using the new PFT

scheme, though both PFT schemes retain some albedo biases related to model structural errors.

Uncertainty from PFT maps is an important and relevant topic to land surface modeling specifically and Earth Science more generally. The specific impact of PFT maps on albedo simulations is highly relevant to studies of global climate, as albedo feedbacks are one of the most important mechanisms for vegetation impacts on regional and global climate, especially at high latitudes. The land surface model (CLASSIC) and the simulation setup appear appropriate for the research questions about the sensitivity of albedo simulations to PFT maps. The description of the land cover and ancillary datasets is thorough, and the data are well-suited to the study objectives. The implementation of mapping these land cover and ancillary datasets onto an improved PFT map is well-described, well-thought-out, and appears robust. The results are clear and compelling, and the conclusions are appropriate to the scope of the results.

**Thank you for your overall positive review of our manuscript.**

I have a few suggestions, primarily related to the paper's organization and presentation.

(1) Most importantly, the exact way that PFT fractions are used in CLASSIC, *especially for the physical calculation of albedo*, needs to be explained more clearly (see detailed comment below).

**Thank you for noting this. We agree that these information will be helpful for better understand the linkage between biases in PFT fractions and simulated albedo by CLASSIC. We will add these when revising our manuscript.**

(2) I found the description of the study's workflow around generating PFT maps (Section 3) confusing and hard to follow; even after multiple reads, I'm not 100% certain exactly what was done or how the pieces fit together. I would suggest adding a more detailed high-level description of what was done at the beginning of Section 3 (the authors should feel free to borrow text from my summary above, assuming it's an accurate reflection of what was done). I would consider a much more detailed version of the flowchart in Figure 2 that indicates exactly which information is flowing where, with reference to the subsections describing that flow of information.

**Thank you for your efforts in summarizing the PFT mapping methods above. It is an accurate reflection of what was done. We will include a more detailed high-level description of the workflow (may borrow some of your text) at the beginning of Section 3**

**and provide a more detailed version of the flowchart in Figure 2 as suggested when revising our manuscript.**

(3) I found the somewhat unorthodox structure of the paper --- where both the methods for PFT mapping and the results thereof (in terms of both land cover distributions and simulated albedo) --- to be confusing. I would suggest having a single methods section clearly focused on how the study was done, and a separate results section that in turn is broken down into (a) differences in land cover and PFT maps between the different approaches, and (b) resulting differences in simulated albedo. Somewhat related to this, I would also only keep details that are directly relevant to this analysis in the methods and move asides and mentions of related work to the discussion (or remove them from the paper altogether). This was especially true of the global maps described in Section 3.3 --I read this section expecting to see global simulations and was surprised to see these absent...which is fine --- they are not necessary to the success of the paper --- but adds confusion to what is already a pretty dense paper.

**The Methods section in the first draft of the manuscript was relatively short, so we combined the Methods and Results into one Section. However, the Methods section has evolved into a considerable length during the revising/finalizing process. We agree that it is better to put the Methods and Results into two sections. Thank you for pointing this out and for your helpful suggestions on separating them.**

**The global maps are described in Section 3.3, which provide a reference to Arora et al. (2022). We agree that it would improve the logical flow of the text if moving this and some other discussion to a separate Discussion section. We will incorporate these excellent suggestions when revising our manuscript**

(4) A minor suggestion: Somewhere in the introduction and/or discussion, it may be worth explicitly distinguishing several categories of approaches for modeling PFTs: (1) Static, where the PFT for a particular pixel is assigned once, exogenously, and persists over the course of the simulation; (2) Forced, where PFTs are still assigned exogenously but can vary through time (e.g., based on scenarios of land cover/land-use change); and (3) Dynamic, where PFTs compete with each other within a pixel through explicitly represented ecological processes (e.g., see the review of vegetation demography models in Fisher et al. 2018 DOI: 10.1111/gcb.13910). I suspect that the relative sensitivity of model results to input PFT maps will vary across these different model types (though I fully expect all of these model types to be sensitive to input PFT maps!).

**Thank you for your suggestion and providing the relevant reference. We agree that it will be helpful to describe explicitly the different approaches for modeling PFTs considering the focus of the manuscript. We will add these when revising our manuscript.**

Overall, I found this to be a well-thought-out and well-executed technical study on an important and relevant topic that is presented in an awkward way. My recommendation is for a significant but almost entirely cosmetic and organizational revision.

Detailed comments:

– – – – – – – – – – – – – – – – – – – – – – – – – – – – – – – – – – – – – – – – – – – – – -

[L190-195]
This is unclear. How does vegetation heterogeneity --- i.e., the four PFTs used for the physics --- represented in the physics scheme? Are the two sub-grid areas with vegetation (with an without snow) in turn a weighted average of parameters from these 4 PFTs? Or is just one PFT selected for the parameterization? Or are parameters for the physics identical? This is especially important to describe clearly and thoroughly because the interpretation of the results hinges primarily on this component.

**In CLASS (the physics module), the albedo of each PFT/vegetation category is calculated separately, first over bare soil and then over a snow pack. The aggregated visible and near-infrared albedos for the bulk canopy are incremented using the current values weighted by the fractional coverage of the vegetation category (Verseghy 1993). We agree that these details will be helpful in understanding the results, which we will add when revising our manuscript.**

[L205-210]
Please clearly indicate which configuration was used in this study --- i.e., was the biogeochemistry on or off? Information about whichever configuration was *not* used in the study is extraneous and can be removed.

**The biogeochemistry was on in the simulations performed in this study. We will clarify this when revising our manuscript.**

---

## Author Comment (AC2)

EGUsphere, referee comment RC2
https://doi.org/10.5194/egusphere-2022-923-RC2, 2023
**Comment on egusphere-2022-923**

Anonymous Referee #2

Referee comment on "Mapping of ESA-CCI land cover data to plant functional types for use in the CLASSIC land model" by Libo Wang et al., EGUsphere, https://doi.org/10.5194/egusphere-2022-923-RC2, 2023

**We thank Referee #2 for their helpful comments. Our replies to his/her comments are shown in bold below.**

Review of "Mapping of ESA-CCI land cover data to plant functional types for use in the CLASSIC land model"

This study focuses on sources of uncertainty in the creation/application of Plant Functional Types (PFTs) in Land Surface Models. The authors highlight the roles of expert judgement and differences in Land Cover (LC) datasets as sources of variation in the distribution and parameterization of PFTs. Focusing on CANADA, the study generates an improved PFT distribution map through the creation of a hybrid LC dataset (combining multiple LC layers) followed by the creation of a new crosswalk table that translates LC to a standard PFT scheme. The study evaluates the influence of this approach using the CLASSIC model to compare simulated winter albedo with new and old PFT representations. The new approach preforms better than model runs based on older PFTs, and the model is evaluated in an interesting sub-pixel PFT composition context.

The motivation for the study is compelling and the crosswalk approach appears to be a tangible improvement to PFT methods.

**Thank you for your overall positive review of our manuscript.**

A primary area of possible improvement for the manuscript is the acknowledgement of its own study limitations in general. A clear example can be found in Section 3.3 where Table 4 is presented. While the rest of the paper centers around the creation of improved PFTs in Canada, it is unclear to what degree this global version is appropriately validated for general use across other regions (e.g., those which were not compared with the LiDAR dataset). This section provides a qualitative and partially anecdotal assessment but evidence for these points is not presented in this manuscript. Likely this could

require a lot to truly validate, so I would suggest bringing in a more explicit acknowledgement of what the actual use-case and limits for this table are. I realize there is some discussion elsewhere in the manuscript regarding global products described in other papers. In general, the limitations of the approach could be explored more.

**Thank you for noting this and for your understanding that it is challenging to truly validate global PFT results presented in Section 3.3 and Table 4, which is beyond the scope of the current study. Considering that our manuscript focuses on Canada (Methods and Results), we agree that it is a bit confusing to have Section 3.3 in the main text as pointed out by Referee #1. We will acknowledge the limitations of the global maps in the main text as suggested, and move Section 3.3 and Table 4 to the Supplement when revising our manuscript.**

A final suggestion would be to be more explicit about each step in the creation of these layers and crosswalk tables. It is often unclear exactly what is done. I will note that the paper is presented in a high level of detail in many places.

**Thank you for your suggestion. We will add more details and describe each step in the creation of the layers and tables more explicitly when revising our manuscript.**

This paper focuses on an important topic for the improvement of Land Surface Models. The manuscript could be improved by acknowledging limitations and by increasing the clarify regarding the details of the methods.

**Thank you for your overall positive review of our manuscript. We will acknowledge the limitations and add more details to the methods section when revising our manuscript.**

Specific

L3 "found" how?

**We will replace it with "Previous studies have shown large differences…"**

L4 "differences arise from the differences" needs an edit

**Thank you for noting this. We will modify it when revising our manuscript.**

L11 What specific study? Maybe it should be "Previous work has shown." Not sure.

**Thank you for noting this. We will use "Previous work has shown." When revising our manuscript.**

L34 It could also be useful to mention (somewhere) what some of the other approaches are beyond PFTs.

**Yes, this point is also raised by Referee #1. Beyond PFTs, species-based models represent vegetation at the level of individual plants and species . These models represent spatial variability in the light environment and simulate competitive exclusion, succession, and coexistence of tree species (Smith et al., 2001). This is computationally expensive, which is often being addressed by limiting the spatial scope and temporal frequency. As a compromise, "cohort-based" models have been developed where individual plants with similar properties**

(size, age, functional type) are grouped together (Fisher et al., 2018). An alternative to the above is trait-based models which focus on the organism traits representing their physiological, morphological, or life-history characteristics (Zakharova et al., 2019). We will include these in the Introduction when revising our manuscript.

Fisher, R. A., Koven, C. D., Anderegg, W. R. L. et al.: Vegetation demographics in Earth System Models: A review of progress and priorities. Glob Change Biol., 24, 35–54, https://doi.org/10.1111/gcb.13910, 2018.

Smith, B., Prentice, I. C., & Sykes, M. T. : Representation of vegetation dynamics in the modelling of terrestrial ecosystems: Comparing two contrasting approaches within European climate space. Global Ecology & Biogeography, 10, 621–637, 2001.

Zakharova, L., Meyer, K. M., Seifan, M.: Trait-based modelling in ecology: A review of two decades of research, Ecological Modelling, 407, https://doi.org/10.1016/j.ecolmodel.2019.05.008, 2019.

L138 So does this mean that "herbs" in VLCE remain herbs if they are not "croplands" in NALCMS?

Yes, "herbs" in VLCE remain herbs if they are not "croplands" in NALCMS. We will clarify this when revising our manuscript.

L141 I appreciate the detailed description of each dataset.
Thank you for noting this.

L218 How was this disaggregation done?

This was done following the methodology by Melton and Arora (2016). Surface temperature, surface pressure, specific humidity, and wind speed are linearly interpolated. Long-wave radiation is uniformly distributed across a 6 h period, and shortwave radiation is diurnally distributed over a day based on a grid cell's latitude and day of year with the maximum value occurring at solar noon. Precipitation is treated following Arora (1997), where the total 6 h precipitation amount is used to determine the number of wet half hours in a 6 h period. The 6 h precipitation amount is then spread randomly, but conservatively, over the wet half-hourly periods. We will add these and the references when revising our manuscript.

Arora, V.: Land surface modelling in general circulation models: a hydrological perspective, PhD thesis, Department of Civil and Environmental Engineering, University of Melbourne, 1997.

Melton, J. R. and Arora, V. K.: Competition between plant functional types in the Canadian Terrestrial Ecosystem Model (CTEM) v. 2.0, Geoscientific Model Development, 9, 323–361, 2016.

L243 This is a particularly important part of the paper but does not feel fully fleshed out. Very little detail is provided for the creation of the tables, and Figure 2 is leaned on heavily. However, Figure 2 doesn't stand alone for several reasons. Acronyms could be spelled out (even simple ones) and some description of the processes being depicted might help.

**Thank you for your suggestions. We will provide more details on the creation of the tables in the text and Figure 2, including spelling out the acronyms and description in the caption of figure 2.**

Table 1 Define the numbers above the PFTs

**We will add the full names of the PFTs to the caption of Table 1.**

Table 1 Why are C3 and C4 grasses combined? You mention separating C3 and C4 using Still et al 2003 (L263) but you also mention combining them because C4 contribution is negligible in Canada (L788). C4 grasses are indeed more common in warmer conditions, but they also do comprise an important part of some grasslands in Canada. It could be useful to define what "negligible" means so that the magnitude of error from this is more explicit. As an example, the percentage of C4 grass species in the regional flora can reach ~24% (C4 Plant Biology 1999). Still et al 2003 is a coarse, global, and physiologically-based estimate.

**Table 1 shows the cross-walking table for mapping the 30 m Hybrid land cover map to CLASSIC PFTs. The Hybrid map does not distinguish C3 from C4 vegetation. The splitting of C3 and C4 is based on the fractional distribution of C4 vegetation in Still and Berry (2003), which is at much lower resolution (1deg). Thus the splitting was done at a later stage when producing the PFTs.**

**Though the main objective of this study is to develop a new cross-walking table over the Canada domain, the ultimate goal is to extend the table to the global scale. It is desirable to split the C3/C4 vegetation based on a global dataset, i.e. Still and Berry (2003). We agree that the resolution of the dataset is rather coarse (1deg), however, we hadn't found a global dataset with finer resolution at the time when carrying out this study.**

**Based on the fractional distribution of C4 vegetation in Still and Berry (2003) and the Hybrid map, the average fraction is 0.5% for C4 crop and 0.1% for C4 grasses in the Canada domain. We will include this information when revising our manuscript.**

L269 It is sometimes unclear exactly what was done, and the LiDAR data are a good example of that. In what way were these data used to inform this partitioning? How well do the LiDAR data align with the other datasets?

**We overlay the Lidar plots on the Hybrid land cover map in ArcMap. Samples (20 to 40, note these classes do not cover large areas in Canada) for the four mixed classes (Sub-polar taiga needleleaf forest, Shrubland, Wetland, and Wetland-treed) in the Hybrid map are selected where there are Lidar plots data. The vegetation coverage data (for canopy height above 2 m) from Lidar plots for samples of each class are used to compute an average coverage of tall vegetation (> 2 m) for that class, which is then used to assign forest fractions for these classes in Table 1. We will add these details when revising our manuscript.**

L311 "cslass"

**Thank you for noting this, we will fix it.**

---

## Author Response (AR1)

Dear Dr. Bond-Lamberty,

We appreciate the thorough review provided by the reviewers. We have revised the manuscript in light of their comments and recommendations. We have attached a copy of the reviewers' comments, followed by our responses. Major changes to the manuscript include:

(1) We have expanded the Introduction by adding the description of other methods used to represent vegetation in models beyond the PFT (Plant Functional Type) method (L32-47). We have also included the different approaches for modeling PFTs in the Introduction (L48-52).

(2) We have included more details about albedo parameterization in CLASSIC in Section 2.4.1 (L213-223) and Appendix A.

(3) We have separated the Methods and Results in the original Section 3 into two Sections: Section 3 (Methods) and Section 4 (Results). A more detailed high-level summary on the PFT mapping methods has been added at the beginning of the new Section 3 (L282-295). More details on the creation of the cross-walking tables have been added in Section 3 and Figure 2.

(4) We have moved the main part of Section 3.3 on the creation of PFTs at the global scale to Appendix B.

(5) We have added a paragraph acknowledging the limitations in this study in Section 5 (L511-529).

We hope that these changes have satisfactorily addressed the comments by the reviewers.

Best Regards,

Libo Wang

**Comment on egusphere-2022-923**
Anonymous Referee #1

Referee comment on "Mapping of ESA-CCI land cover data to plant functional types for use in the CLASSIC land model" by Libo Wang et al., EGUsphere, https://doi.org/10.5194/egusphere-2022-923-RC1, 2022

**We thank Referee #1 for their helpful comments. Our responses to his/her comments are shown in bold below.**

General comments:

This study evaluates the impact of uncertainties and biases in plant functional type (PFT) maps that are used as inputs to land surface models. The specific aim is to quantify the impact of a revised PFT map on winter albedo simulations by the Canadian Land Surface Scheme Including Biogeochemical Cycles (CLASSIC) land surface model. The improved PFT map is generated through a multi-step process that combines multiple land cover maps at different spatial and categorical resolutions with ancillary data on tree cover and vegetation height. First, the authors combine two existing land cover maps (North America Land Change Monitoring System, NALCMS; and Virtual Land Cover Engine, VLCE) to produce a harmonized 30 m land cover map for North America with improved categorical precision (i.e., more precise and accurate category labels). Next, the authors perform a direct mapping of classes from this hybrid land cover map onto the CLASSIC PFT scheme, such that each land cover class corresponds to a particular mix of PFTs as represented in CLASSIC. This step is supported by insights from vegetation height data from an airborne LiDAR campaign over parts of Canada. Next, the authors jointly combine the 30 m hybrid land cover dataset above with a 30 m tree cover fraction dataset (based on the Hansen Landsat analysis) to map sub-pixel fractional composition for classes in the European Space Agency (ESA) Climate Change Initiative (CCI) land cover map (300 m spatial resolution). Next, the authors use this analysis to map the ESA-CCI land cover classes onto PFT mixtures as represented in CLASSIC. Since the ESA-CCI dataset is global, this then allows the authors to perform CLASSIC simulations globally (with some corrections based on exploratory analysis of the resulting PFT map). Finally, the authors perform simulations for Canada and Alaska with the CLASSIC model using its original PFT map (GLC2000) and the revised ESA-CCI scheme described above, specifically looking at differences in simulated winter albedo (which is also compared to the MODIS MCD43C3 white-sky albedo product). Results show that albedo predictions are generally more accurate using the new PFT scheme, though both PFT schemes retain some albedo biases related to model structural errors.

Uncertainty from PFT maps is an important and relevant topic to land surface modeling specifically and Earth Science more generally. The specific impact of PFT maps on albedo simulations is highly relevant to studies of global climate, as albedo feedbacks are one of the most important mechanisms for vegetation impacts on regional and global climate, especially at high latitudes. The land surface model (CLASSIC) and the simulation setup appear appropriate for the research questions about the sensitivity of albedo simulations to PFT maps. The description of the land cover and ancillary datasets is thorough, and the data are well-suited to the study objectives. The implementation of mapping these land cover and ancillary datasets onto an improved PFT map is well-described, wellthought-out, and appears robust. The results are clear and compelling, and the conclusions are appropriate to the scope of the results.

**Thank you for your overall positive review of our manuscript.**

I have a few suggestions, primarily related to the paper's organization and presentation.

(1) Most importantly, the exact way that PFT fractions are used in CLASSIC, *especially for the physical calculation of albedo*, needs to be explained more clearly (see detailed comment below).

**Thank you for noting this. We agree that these information will be helpful for better understand the linkage between biases in PFT fractions and simulated albedo by CLASSIC. We have included more details about the albedo parameterization in CLASSIC in Section 2.4.1 (L213-223) and Appendix A:**

**"The visible and near-infrared albedos of each PFT/vegetation category are specified. These albedos are further modified by taking into account the fraction of the ground that is seen from the sky above referred to as the sky view factor (which is modelled as a function of the leaf area index). The albedo of the ground that is seen from the sky above depends on if the ground is snow covered or not but also on the soil moisture of the top soil layer, since wet soil is darker than the dry soil. Canopy snow processes such as interception/unloading, sublimation, and melt are all simulated. The aggregated visible and near-infrared albedos for the bulk canopy are incremented using the current values weighted by the fractional coverage of the vegetation categories (Verseghy 1993). More details can be found in Appendix A. The overall surface albedo of a grid cell is computed as a weighted mean using the fractional coverages for the four sub-areas."**

(2) I found the description of the study's workflow around generating PFT maps (Section 3) confusing and hard to follow; even after multiple reads, I'm not 100% certain exactly what was done or how the pieces fit together. I would suggest adding a more detailed high-level description of what was done at the beginning of Section 3 (the authors should feel free to borrow text from my summary above, assuming it's an accurate reflection of what was done). I would consider a much more detailed version of the flowchart in Figure 2 that indicates exactly which information is flowing where, with reference to the subsections describing that flow of information.

**Thank you for your efforts in summarizing the PFT mapping methods above. It is an accurate reflection of what was done. We have included a more detailed high-level description of the workflow (borrowed some of your text) at the beginning of Section 3 (L282-295):**

**"The CW-table for the ESA-CCI LC dataset is generated through a multi-step process that combines multiple land cover maps at different spatial and categorical resolutions with ancillary data on tree cover and vegetation height. This includes the following steps: (1) combining two existing land cover maps (NALCMS and VLCE) to produce a harmonized 30 m land cover (Hybrid) map with improved categorical precision (as described in Section 2.1); (2) creating a CW-table for the Hybrid land cover map through a direct mapping of classes from the Hybrid map onto the CLASSIC PFTs, such that each land cover class corresponds to a particular mix of PFTs as represented in CLASSIC. This step is supported by vegetation height data from an airborne Lidar campaign over parts of Canada; (3) computing the sub-pixel fractional composition for classes in the ESA-CCI land cover map (300 m resolution) based on the 30 m Hybrid land cover dataset and the Hansen tree cover fraction dataset; (4) using the sub-pixel fractional composition analysis to create a CW-table for mapping the ESA-CCI land cover classes onto PFTs as represented in CLASSIC; and (5) since the ESA-CCI dataset is global, the CW-table developed over Canada is extended to the whole globe."**

**More details on the creation of the cross-walking tables have been added throughout Section 3 and in the flowchart in Figure 2.**

(3) I found the somewhat unorthodox structure of the paper --- where both the methods for PFT mapping and the results thereof (in terms of both land cover distributions and simulated albedo) --- to be confusing. I would suggest having a single methods section clearly focused on how the study was done, and a separate results section that in turn is broken down into (a) differences in land cover and PFT maps between the different approaches, and (b) resulting differences in simulated albedo. Somewhat related to this, I would also only keep details that are directly relevant to this analysis in the methods and move asides and mentions of related work to the discussion (or remove them from the paper altogether). This was especially true of the global maps described in Section 3.3 --I read this section expecting to see global simulations and was surprised to see these absent...which is fine --- they are not necessary to the success of the paper --- but adds confusion to what is already a pretty dense paper.

**Thank you for pointing this out and for your helpful suggestions on separating Section3. We have separated the Methods and Results in the original Section 3 into two Sections. We have moved the main part of Section 3.3 on the creation of PFTs at the global scale to Appendix B.**

(4) A minor suggestion: Somewhere in the introduction and/or discussion, it may be worth explicitly distinguishing several categories of approaches for modeling PFTs: (1) Static, where the PFT for a particular pixel is assigned once, exogenously, and persists over the course of the simulation; (2) Forced, where PFTs are still assigned exogenously but can vary through time (e.g., based on scenarios of land cover/land-use change); and (3) Dynamic, where PFTs compete with each other within a pixel through explicitly represented ecological processes (e.g., see the review of vegetation demography models in Fisher et al. 2018 DOI: 10.1111/gcb.13910). I suspect that the relative sensitivity of model results to input PFT maps will vary across these different model types (though I fully expect all of these model types to be sensitive to input PFT maps!).

**Thank you for your suggestion and providing the relevant reference. We have included the different approaches for modeling PFTs in the Introduction (L48-52): "There are three approaches for modeling PFTs: (1) static, where the fractional coverage of PFTs is prescribed and does not vary through time; (2) forced, where the fractional coverage of PFTs is still prescribed but vary through time based on scenarios of land cover/land-use change; and (3) dynamic, where the fractional coverage of PFTs is simulated dynamically with competition for available space and resources between PFTs (Fisher et al., 2018; Melton and Arora, 2016)."**

**Fisher, R. A., Koven, C. D., Anderegg, W. R. L. et al.: Vegetation demographics in Earth System Models: A review of progress and priorities. Glob Change Biol., 24, 35–54, https://doi.org/10.1111/gcb.13910, 2018.**

Overall, I found this to be a well-thought-out and well-executed technical study on an important and relevant topic that is presented in an awkward way. My recommendation is for a significant but almost entirely cosmetic and organizational revision.

Detailed comments:

[L190-195]
This is unclear. How does vegetation heterogeneity --- i.e., the four PFTs used for the physics --- represented in the physics scheme? Are the two sub-grid areas with vegetation (with an without snow) in turn a weighted average of parameters from these 4 PFTs? Or is just one PFT selected for the parameterization? Or are parameters for the physics identical? This is especially important to describe clearly and thoroughly because the interpretation of the results hinges primarily on this component.

**We have included more details on the albedo parameterization in CLASSIC in Section 2.4.1 (L213-223) and Appendix A. Please see our detailed response above.**

[L205-210]

Please clearly indicate which configuration was used in this study --- i.e., was the biogeochemistry on or off? Information about whichever configuration was \*not\* used in the study is extraneous and can be removed.

**The biogeochemistry was on in the simulations performed in this study. We have clarified this in Section 2.4.1.**

Comment on egusphere-2022-923
Anonymous Referee #2

Referee comment on "Mapping of ESA-CCI land cover data to plant functional types for use in the CLASSIC land model" by Libo Wang et al., EGUsphere, https://doi.org/10.5194/egusphere-2022-923-RC2, 2023

**We thank Referee #2 for their helpful comments. Our responses to his/her comments are shown in bold below.**

Review of "Mapping of ESA-CCI land cover data to plant functional types for use in the CLASSIC land model"

This study focuses on sources of uncertainty in the creation/application of Plant Functional Types (PFTs) in Land Surface Models. The authors highlight the roles of expert judgement and differences in Land Cover (LC) datasets as sources of variation in the distribution and parameterization of PFTs. Focusing on CANADA, the study generates an improved PFT distribution map through the creation of a hybrid LC dataset (combining multiple LC layers) followed by the creation of a new crosswalk table that translates LC to a standard PFT scheme. The study evaluates the influence of this approach using the CLASSIC model to compare simulated winter albedo with new and old PFT representations. The new approach preforms better than model runs based on older PFTs, and the model is evaluated in an interesting sub-pixel PFT composition context.

The motivation for the study is compelling and the crosswalk approach appears to be a tangible improvement to PFT methods.

**Thank you for your overall positive review of our manuscript.**

A primary area of possible improvement for the manuscript is the acknowledgement of its own study limitations in general. A clear example can be found in Section 3.3 where Table 4 is presented. While the rest of the paper centers around the creation of improved PFTs in Canada, it is unclear to what degree this global version is appropriately validated for general use across other regions (e.g., those which were not compared with the LiDAR dataset). This section provides a qualitative and partially anecdotal assessment but evidence for these points is not presented in this manuscript. Likely this could require a lot to truly validate, so I would suggest bringing in a more explicit acknowledgement of what the actual use-case and limits for this table are. I realize there is some discussion elsewhere in the manuscript regarding global products described in other papers. In general, the limitations of the approach could be explored more.

**Thank you for noting this and for your understanding that it is challenging to truly validate global PFT results presented in Section 3.3 and Table 4, which is beyond the scope of the current study. Considering that our manuscript focuses on Canada (Methods and Results), we agree that it is a bit confusing to have Section 3.3 in the main text as also pointed out by Referee #1.**

**We have moved the main part of Section 3.3 on the creation of PFTs at the global scale to Appendix B. We have added a paragraph acknowledging the limitations in this study in Section 5 (L511-529):**

**"Our PFT mapping approach for the ESA-CCI dataset is mainly based on sub-pixel fractional composition analyses using the Hybrid map and the Hansen tree cover fraction data, and therefore the accuracy of the latter two datasets affects the PFT mapping process. Some LC categories in the ESA-CCI legend either have limited presence or no presence in Canada, such as the Needleleaf deciduous trees, Broadleaf Evergreen trees, and Broadleaf Dry Deciduous trees etc., and the sub-pixel fractional composition analyses therefore can not be performed for these LC categories. The needleleaf deciduous tree cover classes are assigned to the same fractions as the needleleaf evergreen tree cover classes in the CW-table, and values based on the default CW-table from the ESA-CCI user guide are used for the other LC categories. Therefore potentially large uncertainties may be associated with these classes in the resulting fractional coverage of PFTs especially at the global scale. Similar analyses for other regions (e.g. Eurasia and tropics) for which high quality regional land cover maps are available will be helpful in reducing these uncertainties in the future work. In addition, the exercise of mapping PFTs at the global scale in this study reveals that there are inconsistencies in the representation of fractional coverage for some LC categories in the ESA-CCI map for different regions of the globe. Future improvements in the consistency of the LC categories globally in the ESA-CCI LC product would greatly benefit the land surface and the earth system modelling community. In the meantime, caution should be exercised when using this product for mapping PFTs represented in any LSM based on a single cross-walking table at the global scale."**

A final suggestion would be to be more explicit about each step in the creation of these layers and crosswalk tables. It is often unclear exactly what is done. I will note that the paper is presented in a high level of detail in many places.

**Thank you for your suggestion. We have included a more detailed high-level description of the workflow at the beginning of Section 3 (L282-295). More details on the creation of the cross-walking tables have been added throughout Section 3 and the flowchart in Figure 2.**

This paper focuses on an important topic for the improvement of Land Surface Models. The manuscript could be improved by acknowledging limitations and by increasing the clarify regarding the details of the methods.

**Thank you for your overall positive review of our manuscript. We have added a paragraph acknowledging the limitations in this study and included more details of the methods. Please see our detailed responses above.**

Specific

L3 "found" how?

**We have replaced "found" with "Previous studies have shown …"**

L4 "differences arise from the differences" needs an edit

**Thank you for noting this. We have modified the sentence to the following:**

**"Previous studies have shown large differences in the geographical distribution of PFTs currently used in various LSMs, which may arise from the differences in the underlying land cover products but also the methods used to map or reclassify land cover data to the PFTs that a given LSM represents."**

L11 What specific study? Maybe it should be "Previous work has shown." Not sure.

**Thank you for noting this. We have now used "Previous work has shown" as suggested.**

L34 It could also be useful to mention (somewhere) what some of the other approaches are beyond PFTs.

**We have expanded the Introduction by adding the description of other methods used to represent vegetation in models beyond the PFT method (L32-47):**

**"Plant functional types (PFTs) are groups of plant species that share similar structural, phonological, and physiological traits, and have been commonly used in LSMs to represent vegetation distribution. This simplification has allowed the simulation of structural attributes of vegetation dynamically within ESMs (Arora & Boer, 2010; Bonan et al., 2003; Krinner et al., 2005). In order to improve the representation of ecosystem ecology and vegetation demographic processes within ESMs, both species-based and trait-based models have been attempted in LSMs (Fisher et al., 2018; Zakharova et al., 2019). However, these individual-based models are computationally too expensive to model biogeochemical processes, especially photosynthesis and the carbon cycle at the global scale (Bonan et al., 2002; Smith et al., 1997; 2001). As a compromise, "cohort-based" models have been developed where individual plants with similar properties (size, age, functional type) are grouped together and have been implemented in some ESMs (Fisher et al., 2018). Though there are limitations in PFTs-based models (Scheiter et al., 2013; Zakharova et al., 2019), PFTs are commonly used in LSMs that participate routinely in the Global Carbon Project (Friedlingstein et al., 2020) and in ESMs that participate in the Coupled Models Intercomparison Project (CMIP, Wang et al., 2016)."**

**Bonan, G. B., Levis, S., Sitch, S., Vertenstein, M., and Oleson, K. W.: A dynamic global vegetation model for use with climate models: Concepts and description of simulated vegetation dynamics. Global Change Biology, 9, 1543–1566, 2003.**

**Krinner, G., Viovy, N., de Noblet-Ducoudre, N., Ogee, J., Polcher, J., Friedlingstein, P. et al.: A dynamic global vegetation model for studies of the coupled atmosphere-biosphere system, Global Biogeochemical Cycles, 19(1), GB1015, 2005.**

**Scheiter, S., Langan, L., Higgins, S.I.: Next-generation dynamic global vegetation models: learning from community ecology, New Phytol., 198, 957–969, https://doi.org/10.1111/nph.12210, 2013.**

**Smith, B., Prentice, I. C., & Sykes, M. T. : Representation of vegetation dynamics in the modelling of terrestrial ecosystems: Comparing two contrasting approaches within European climate space. Global Ecology & Biogeography, 10, 621–637, 2001.**

**Zakharova, L., Meyer, K. M., Seifan, M.: Trait-based modelling in ecology: A review of two decades of research, Ecological Modelling, 407, https://doi.org/10.1016/j.ecolmodel.2019.05.008, 2019.**

L138 So does this mean that "herbs" in VLCE remain herbs if they are not "croplands" in NALCMS?

**Yes, "herbs" in VLCE remain herbs if they are not "croplands" in NALCMS. We have clarified this in the revised manuscript (L155).**

L141 I appreciate the detailed description of each dataset.
**Thank you for noting this.**

L218 How was this disaggregation done?

**We have added the following in Section 2.4.2 (L244-255):**

**"The 6-hourly data are disaggregated on-the-fly within CLASSIC into half-hourly data following the methodology by Melton and Arora (2016) for the following seven meteorological variables that are used to force the model: 2 m air temperature, total precipitation, specific humidity, downward solar radiation flux, downward longwave radiation flux, surface pressure, and wind speed. Surface temperature, surface pressure, specific humidity, and wind speed are linearly interpolated. Long-wave radiation is uniformly distributed across a 6-hour period, and shortwave radiation is diurnally distributed over a day based on a grid cell's latitude and day of year with the maximum value occurring at solar noon. Precipitation is treated following Arora (1997), where the total 6-hour precipitation amount is used to determine the number of wet half hours in a 6-hour period. The 6-hour precipitation amount is then spread randomly, but conservatively, over the wet half-hourly periods."**

**Arora, V.: Land surface modelling in general circulation models: a hydrological perspective, PhD thesis, Department of Civil and Environmental Engineering, University of Melbourne, 1997.**

L243 This is a particularly important part of the paper but does not feel fully fleshed out. Very little detail is provided for the creation of the tables, and Figure 2 is leaned on heavily. However, Figure 2 doesn't stand alone for several reasons. Acronyms could be spelled out (even simple ones) and some description of the processes being depicted might help.

**Thank you for your suggestions. We have added more details on the creation of the cross-walking tables throughout Section 3 and in Figure 2. We have included a more detailed high-level description of the workflow (describing the steps in Figure 2) at the beginning of Section 3 (L282-295).**

Table 1 Define the numbers above the PFTs

**We have included the full names of the PFTs in the caption of Table 1.**

Table 1 Why are C3 and C4 grasses combined? You mention separating C3 and C4 using Still et al 2003 (L263) but you also mention combining them because C4 contribution is negligible in Canada (L788). C4 grasses are indeed more common in warmer conditions, but they also do comprise an important part of some grasslands in Canada. It could be useful to define what "negligible" means so that the magnitude of error from this is more explicit. As an example, the percentage of C4 grass species in the regional flora can reach ~24% (C4 Plant Biology 1999). Still et al 2003 is a coarse, global, and physiologically-based estimate.

**Table 1 shows the cross-walking table for mapping the 30 m Hybrid land cover map to CLASSIC PFTs. The Hybrid map does not distinguish C3 from C4 vegetation. The separation of C3 and C4 is based on the fractional distribution of C4 vegetation in Still and Berry (2003), which is at much lower resolution (1deg). Thus the separation was done at a later stage when producing the PFTs.**

**Though the main objective of this study is to develop a new cross-walking table over the Canada domain, the ultimate goal is to extend the table to the global scale. It is desirable to split the C3/C4 vegetation based on a global dataset, i.e. Still and Berry (2003). We agree that the resolution of the dataset is rather coarse (1deg), however, we hadn't found a global dataset with finer resolution at the time when carrying out this study.**

**Based on the fractional distribution of C4 vegetation in Still and Berry (2003) and the Hybrid map, the average fraction is 0.5% for C4 crops and 0.1% for C4 grasses in Canada. We have included this information in Section 4.1 (L440-442).**

L269 It is sometimes unclear exactly what was done, and the LiDAR data are a good example of that. In what way were these data used to inform this partitioning? How well do the LiDAR data align with the other datasets?

**We have included more details on the creation of the cross-walking tables in Section 3. The following sentences have been added to explain the use of the LiDAR data (L319-324): "We overlay the Lidar plots on the Hybrid land cover map in ArcGIS. Samples (20 to 40, note that these classes do not cover large areas in Canada) for the four mixed classes in the Hybrid map are selected where there are Lidar data. The vegetation coverage data (for canopy height above 2 m) from Lidar plots for samples of each class are used to compute an average coverage of tall vegetation (> 2 m) for that class, which is then used to assign forest fractions for these four classes in Table 1."**

L311 "cslass"

**Fixed.**

---

## Author Response (AR2)

Dear Dr. Bond-Lamberty,

We have corrected the typos pointed out by the referee and added the following sentences (L443-446) to address the referee's suggestion on highlighting the limitation of the C4 maps used in the study:

"The C4 fraction product from Still and Berry (2003) is available at a much coarser spatial resolution (1°) than other land cover products used in this study, and it is a global product. As such then, the estimated C4 fractions for crops/grasses in Canada used here may not completely agree with those from regional estimates."

We hope that these changes have satisfactorily addressed the comments by the referee.

Best Regards,

Libo Wang